# Identifying plant genes shaping microbiota composition in the barley rhizosphere

Carmen Escudero-Martinez[1,11], Max Coulter[1,2,11], Rodrigo Alegria Terrazas[1,3], Alexandre Foito[4], Rumana Kapadia[1], Laura Pietrangelo[1,5], Mauro Maver [1,6,7], Rajiv Sharma[8], Alessio Aprile [1,9], Jenny Morris[4], Pete E. Hedley[4], Andreas Maurer [10], Klaus Pillen [10], Gino Naclerio[5], Tanja Mimmo [6,7], Geoffrey J. Barton [2], Robbie Waugh [1,4], James Abbott[2] & Davide Bulgarelli [1✉]

A prerequisite to exploiting soil microbes for sustainable crop production is the identification of the plant genes shaping microbiota composition in the rhizosphere, the interface between roots and soil. Here, we use metagenomics information as an external quantitative phenotype to map the host genetic determinants of the rhizosphere microbiota in wild and domesticated genotypes of barley, the fourth most cultivated cereal globally. We identify a small number of loci with a major effect on the composition of rhizosphere communities. One of those, designated the *QRMC-3HS*, emerges as a major determinant of microbiota composition. We subject soil-grown sibling lines harbouring contrasting alleles at *QRMC-3HS* and hosting contrasting microbiotas to comparative root RNA-seq profiling. This allows us to identify three primary candidate genes, including a Nucleotide-Binding-Leucine-Rich-Repeat (*NLR*) gene in a region of structural variation of the barley genome. Our results provide insights into the footprint of crop improvement on the plant's capacity of shaping rhizosphere microbes.

[1] University of Dundee, Plant Sciences, School of Life Sciences, Dundee, UK. [2] University of Dundee, Computational Biology, School of Life Sciences, Dundee, UK. [3] Mohammed VI Polytechnic University, Agrobiosciences Program, Plant & Soil Microbiome Subprogram, Bengurir, Morocco. [4] The James Hutton Institute, Invergowrie, UK. [5] Department of Biosciences and Territory, University of Molise, Campobasso, Italy. [6] Faculty of Science and Technology, Free University of Bozen-Bolzano, Bolzano, Italy. [7] Competence Centre for Plant Health, Free University of Bozen-Bolzano, Bolzano, Italy. [8] Scotland's Rural College, Edinburgh, UK. [9] Department of Biological and Environmental Sciences and Technologies, University of Salento, Lecce, Italy. [10] Institute of Agricultural and Nutritional Sciences, Martin-Luther-University, Halle-Wittenberg, Germany. [11] These authors contributed equally: Carmen Escudero-Martinez, Max Coulter. ✉email: d.bulgarelli@dundee.ac.uk

Plants thrive in association with diverse microbial communities, collectively referred to as the plant microbiota. This is capable of impacting the growth, development and health of their hosts[1–4]. The rhizosphere, the interface between the roots and soil[5], is a key microhabitat for the plant microbiota. For instance, similar to probiotics of the microbiota populating the digestive tract of vertebrates[6], microbes inhabiting the rhizosphere can promote plant growth by facilitating mineral nutrient uptake and pathogen protection[1,7–10].

These interactions do not represent stochastic events but are controlled, at least in part, by the plant genome[11,12]. Resolving the host genetic control of the microbes thriving at the root-soil interface therefore represents one of the prerequisites for the rational manipulation of the plant microbiota for agriculture[13]. This is particularly relevant for the microbiota associated with crop wild relatives, which, having evolved under marginal soil conditions, may represent an untapped resource for low-input agriculture[14,15]. However, despite a footprint of domestication and crop selection having been identified in the taxonomic composition of the rhizosphere microbiota in multiple plant species[16–23], host genes underpinning this diversification remain poorly understood.

Barley is the fourth-most cultivated cereal globally[24] and an attractive experimental model to study plant-microbe interactions in the light of domestication and crop selection. We previously demonstrated that wild genotypes and 'elite' cultivated varieties host contrasting rhizosphere microbiotas[25,26]. In this work, capitalising on an experimental population between barley genotypes at opposing ends of the domestication framework[27] and utilising state-of-the-art genomic[28] and transcriptomic[29] resources, we map host genetic determinants of microbiota composition in the rhizosphere. We identify candidate genes putatively underpinning this trait and define genetic variation occurring at those genes.

## Results

**The composition of the bacterial microbiota in the barley rhizosphere appears to be controlled by a limited number of loci.** We grew 52 genotypes of the progeny of a segregating population between the elite cultivar (*Hordeum vulgare ssp. vulgare*) 'Barke' and the wild ancestor accession (*Hordeum vulgare ssp. spontaneum*) called HID-144 (see the 'Methods' section), hereafter designated 'elite' and 'wild' respectively, in a soil previously used for investigating the barley rhizosphere microbiota[26,30,31] under controlled environmental conditions (see 'Methods'). At early stem elongation (Supplementary Fig. 1), plants were removed from their substrates, and the rhizosphere fractions alongside bulk soil controls were subjected to an amplicon sequencing survey of the 16S rRNA gene.

Having defined a threshold for PCR reproducibility of individual amplicon sequencing variants (ASVs) (Supplementary Fig. 2), representing the terminal taxonomic nodes in our sequencing profiling, we inspected the impact of the parental lines on the composition of the bacterial microbiota. This allowed us to identify 36 ASVs discriminating between the parental lines (Wald test, individual P-value < 0.05, FDR corrected, Fig. 1a). Of these taxa, 27 ASVs, representing 5.39% of the reads, were enriched in and discriminated between the wild genotype and the elite variety (Wald test, individual P-values < 0.05, FDR corrected, Fig. 1a). Conversely, 9 ASVs representing 2.74% of the reads were enriched in and discriminated between the elite variety and the wild genotype (Wald test, individual P-value < 0.05, FDR corrected, Fig. 1a). This differential microbial enrichment between the parental lines was associated with a taxonomic diversification at phylum level: while the wild-enriched profiles

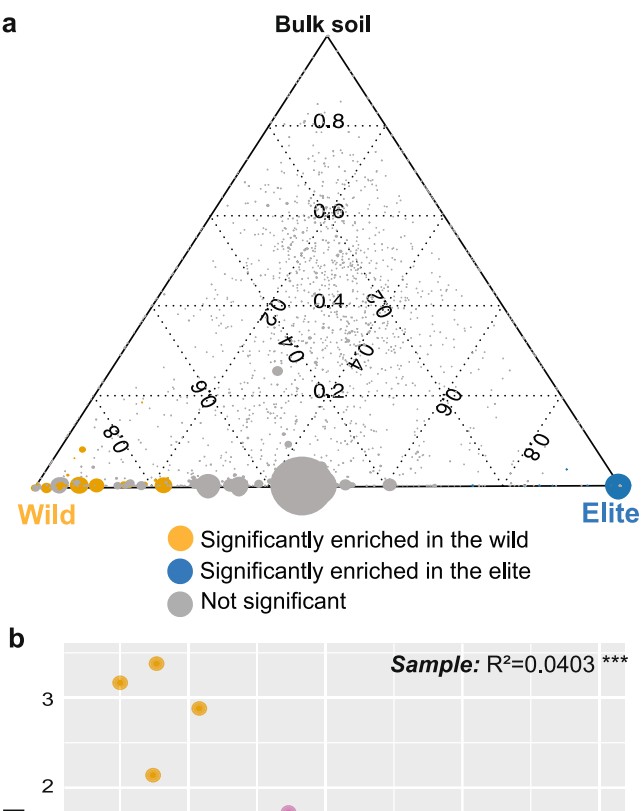

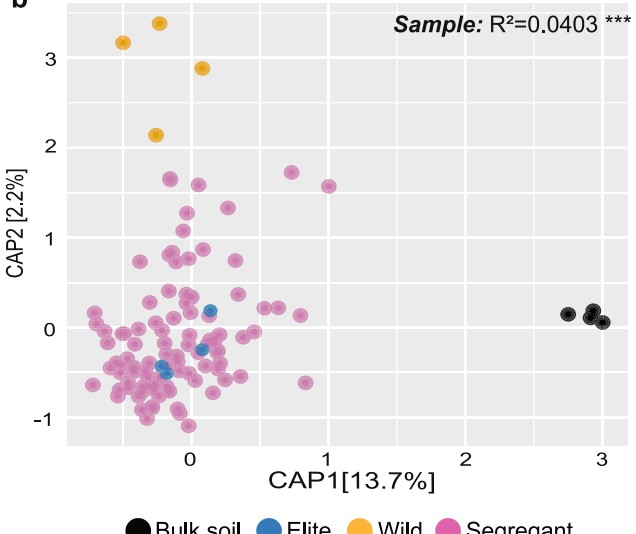

**Fig. 1 Barley microbiota composition displays a quantitative variation in a segregating population between wild and elite parental lines. a** Ternary plot depicting microbiota composition in the elite and wild genotypes as well as bulk soil samples. Each dot illustrates an individual ASV; the size of the dots is proportional to ASV's abundance while their position reflects the microhabitat where bacteria were predominantly identified. Individual dots are colour-coded according to their significant enrichment in the rhizosphere of either parental line (Wald Test, Individual P-values < 0.05, FDR corrected). **b** Canonical Analysis of Principal Coordinates computed on Bray–Curtis dissimilarity matrix. Individual dots in the plot denote individual biological replicates whose colours depict sample type in the bottom part of the figure. The number in the plot depicts the proportion of variance ($R^2$) explained by the factor 'Sample' within the rhizosphere microhabitat, i.e., Elite, Wild or Segregant. The asterisks associated to the $R^2$ value denote its significance, P-value 'Sample' = 0.001; Adonis test, F = 2.23, 5000 permutations. Source data are provided as a Source data file.

encompassed several ASVs classified as Bacteroidota (33.9% of the enriched reads), Firmicutes (32.8%), Proteobacteria (27.9%) Acidobacteria (3.7%) and Myxococcota (1.3%); the elite parent enriched predominantly for members of the phylum Actinobacteria (55.8%), followed by members of Firmicutes (17.8%), Proteobacteria (15.3%), Bacteroidota (10.8%) and Acidobacteria

(0.3%) (Supplementary Fig. 3). Next, we extended our survey to the entire segregating population. We made two observations. First, and consistent with previous investigations in the same reference soil[26,30,31], rhizosphere communities were significantly different from unplanted soil controls as illustrated by sample separation along the x-axis of the Canonical Analysis of Principal coordinates (CAP) (Adonis test between bulk soil and rhizosphere samples, F = 7.49, P-value = 0.001, 5000 permutations, Fig. 1b). Despite identifying a significant impact of genotype on the composition of the rhizosphere samples, we failed to partition this variation into discrete classes (Adonis test, $R^2$ genotype among rhizosphere samples = 0.0403, F = 2.23, P-value = 0.001, 5000 permutations, Fig. 1b). Thus, while samples corresponding to the elite genotype segregated from the wild genotype along the y-axis accounting for the second source of variation in a constrained ordination, individual segregants were distributed between the parental lines. This distribution mirrored the increased proportion of elite genome expected in the original back-crossed $BC_1S_3$ population, with the majority of microbiota profiles of segregating individuals located spatially closer to the elite genotype (Fig. 1b). These observations suggest that microbiota variation in the barley rhizosphere can be used as a trait in quantitative genetic studies.

To gain insights into the host genetic control of the rhizosphere microbiota, we developed a reductionistic approach whereby we used taxa that were differentially recruited between the parental lines and their abundances in the segregating population as quantitative phenotypes to search for significant associations with genetic markers located throughout the barley genome. To ascertain the phylogenetic congruence of the observed microbial trait we repeated this analysis at different taxonomic levels with sequencing reads agglomerated at genus and family level, respectively. For several bacteria we had previously characterised as being differentially abundant between the parental lines, we identified significant associations with individual homozygous or heterozygous alleles at multiple loci across the barley genome. These associations are supported either by marker regression or by a minimum LOD score of 3.43 at ASV, 3.56 at genus and 3.65 at family levels based on a LOD genome-wide significance threshold (alpha level = 0.2; 1000 permutations) (Fig. 2, Supplementary Figs. 4–6, Supplementary Tables 1 and 2, Supplementary Data 1). However, one locus that mapped between 38.7 and 40.6 centimorgans (cM) on chromosome 3H was associated with the recruitment of phylogenetically unrelated bacteria at multiple taxonomic levels. The locus was identified as *QRMC.BaH144-3HS* where *QRMC* stands for QTL–Rhizosphere Microbiota Composition, *BaH144* corresponds to the cross Barke x HID-144 and *3HS* the short arm of chromosome 3H, hereafter referred to as *QRMC-3HS*. All the bacteria recruited at *QRMC-3HS* were significantly enriched in the wild parent. We observed up to four unrelated ASVs representing 5.68% of the enriched bacterial reads in the parental lines were linked to the *QRMC-3HS* classified as *Variovorax sp.*, *Holophaga sp.*, *Sorangium sp.* and *Tahibacter sp.* When taxa were agglomerated at the genus level, the number of significant associations increased to five, including the genus Rhodanobacter. The same analysis computed at family level revealed a congruent phylogenetic pattern associated with this locus represented by Comamonadaceae (the family of the genus Variovorax), Holophagaceae (Holophaga), Polyangiaceae (Sorangium) and Rhodanobacteraceae (Rhodanobacter and Tahibacter) (Supplementary Tables 1–5). *QRMC-3HS* was the only QTL recurrently found at different taxonomic levels presenting associations with up to five taxa across analyses with a LOD threshold established with more stringent criteria (alpha level = 0.05; 1000 permutations) (Supplementary Table 2), and explaining at least ~20% of the phenotypic variance (i.e., sequencing

reads) for the individual taxa significantly associated to it (Supplementary Tables 3–5). These results indicate that *QRMC-3HS* represents a major plant genetic determinant of microbiota recruitment in the barley rhizosphere.

**Wild introgressions at *QRMC-3HS* are associated with compositional changes in the rhizosphere bacterial microbiota.** To validate the results of the mapping exercise, we tested whether barley lines harbouring contrasting alleles, i.e., either 'elite' or 'wild', at *QRMC-3HS* would be associated with distinct microbial phenotypes. We generated two sibling lines designated 124_17, carrying 'elite' alleles at *QRMC-3HS*, and 124_52, harbouring 'wild' alleles at *QRMC-3HS* (Supplementary Fig. 1) by selfing the progeny of line HEB_15_124 which we identified as being heterozygous at the locus of interest (see 'Methods'). Besides the genetic differences at *QRMC-3HS*, the derived lines share 95.5% and 93.3% of their genomes, respectively, with the elite parent based on molecular marker profiling using the barley 50k iSelect SNP Array[32] (Supplementary Table 6, Supplementary Data 3). They also represent bona fide progenies of the population investigated in this study (Supplementary Table 6).

We grew these sibling lines, along with the elite genotype and bulk soil controls, in the same experimental set-up described for the mapping experiment. We quantified 16S rRNA gene total abundance for these rhizosphere and bulk soil samples as a first step towards a comparative microbiota profiling of the new material. This quantification showed no statistical differences of 16S rRNA gene total abundance among the tested genotypes (Kruskal–Wallis test, $\chi^2 = 12.47$, and Dunn's test, P-value < 0.05, Supplementary Fig. 7a). We next inspected three ecological indices of alpha diversity, i.e., within sample microbial diversity, namely 'observed ASVs', 'Chao1' and 'Shannon' indices, proxies for microbiota richness and evenness. This analysis did not reveal significant differences between the communities inhabiting the rhizosphere of the sibling lines and those of the elite Barke at the threshold we imposed (ANOVA and Tukey post hoc test, P-value < 0.05, Supplementary Fig. 8). Conversely, we did identify a significant host-genotype component when we inspected beta-diversity, which is the between sample microbial diversity, a proxy for microbiota diversification (Adonis test, $R^2$ genotype = 0.119, F = 1.84, P-value = 0.005, 5000 permutations). This was manifested by the separation of the communities associated with the tested genotypes, in particular those of line 124_52 (wild-like) from those of the parental line Barke along the axis accounting for the major variation in a CAP (Fig. 3a). We were, however, unable to determine which individual ASVs were responsible for the observed differentiation at the threshold imposed in the mapping experiment (Wald test, individual P-value < 0.05, FDR corrected).

These results nevertheless indicate that genetic variation at *QRMC-3HS* is associated with a significant shift in community composition in the rhizosphere. This trait is not driven by 16S rRNA gene total abundance nor by differences in community richness and evenness. Despite a significant change in community composition, a wild introgression at *QRMC-3HS* is not however sufficient to trigger differential enrichments of individual bacteria.

**Genetic variation at *QRMC-3HS* does not perturb the composition of the barley fungal microbiota.** To investigate whether *QRMC-3HS* could shape the fungal communities inhabiting the rhizosphere, we carried out a similar sequencing survey using the rRNA ITS region. In common with the observed results for the bacterial counterpart, the evaluation of the ITS region total abundance did not reveal significant differences between the sibling lines and the elite parental line Barke (Kruskal–Wallis $\chi^2 = 25.986$, and Dunn's test, individual P-value < 0.05, Supplementary Fig. 7b).

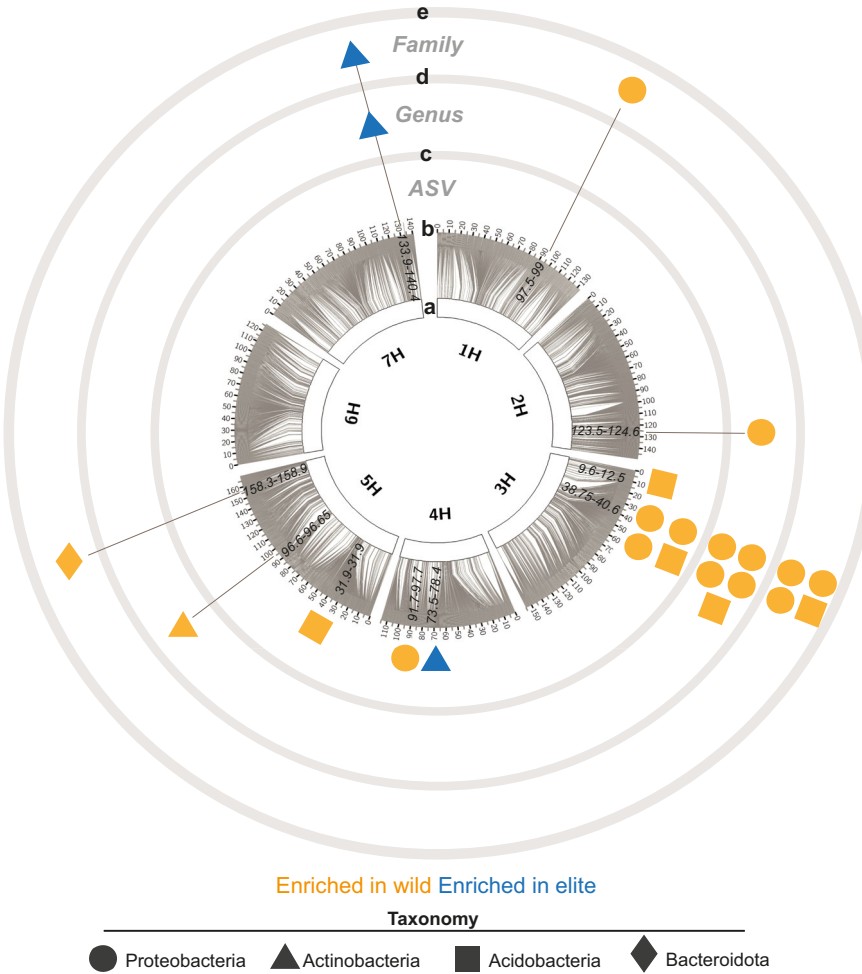

**Fig. 2 Genetic map of the barley determinants of individual bacterial members of the rhizosphere microbiota.** Circos plot depicting **a** the seven barley chromosomes and **b** grey connector lines link the physical position of SNPs with the genetic position in cM as indicated in the outer part of the ring; numbers in black within the individual chromosome define genetic positions (cM) significantly associated (using the function *scanone* implementing interval mapping with a single-QTL model, expectation-maximization algorithm, LOD genome-wide significance threshold 20% adjusted per taxa, 1000 permutations) to the differential enrichment of individual **c** ASVs, **d** genus or **e** family, respectively. Different shapes depict taxonomic assignment at phylum level. Shapes are colour-coded according to the microbiota of the parental line where individual taxa were identified. Source data are provided as a Source data file.

Next, we generated an amplicon sequencing library using primers targeting the rRNA ITS region and identified a total of 216 fungal ASVs after applying filtering criteria (see 'Methods'). When we implemented a beta-diversity analysis of the ITS library, we failed to identify a significant effect of the host genotype on these communities (Adonis test, F = 0.26, *P*-value = 0.963, 5000 permutations). This was further manifested by the lack of spatial separation among microbiota samples of different genotypes in a CAP (Fig. 3b) Likewise, no differentially recruited ASVs were identified in pair-wise comparisons using DESeq2 (Wald test, individual *P*-values < 0.05, FDR corrected).

These observations suggest that the selection pressure exerted by *QRMC-3HS* on the barley microbiota is predominantly confined to its bacterial members.

**QRMC-3HS does not impact other root and yield traits.** To gain mechanistic insights into the plant traits associated with micro-biota diversification, we examined the root macro architecture, as morphological differences in barley roots can alter microbial composition in the rhizosphere[18,30]. When we measured root weight and nine different root morphology parameters of plants grown in the same soil used for microbiota characterisation, no significant differences were found between the sibling lines and the elite genotype at the imposed threshold (i.e., Kruskal–Wallis and post hoc Dunn's test, *P*-values < 0.05, Supplementary Table 7).

Next, we grew the sibling lines and the elite genotype in sterile sand and determined the elemental composition of carbon and nitrogen in their exudates, as both of these elements represent another possible driver of microbial recruitment in the rhizosphere[33–35]. We selected two different timepoints: 2- and 3-weeks post-transplantation, to study the patterns of exudation. The former timepoint is critical for the establishment of the bacterial community in cereals[36], while the latter corresponds to the onset of stem elongation when the rhizosphere is harvested for microbial profiling[26,30,31]. As plants were supplemented with a nutrient solution (see 'Methods'), we used wash-through from unplanted pots as controls. The carbon content was significantly higher in the planted samples compared with the unplanted ones regardless of the timepoint. For instance, unplanted samples wash-through contained just 1.5–3.6% in carbon content weight (w/w), while that of plant exudates was 23–31% in weight, although no significant differences among genotypes were identified at the

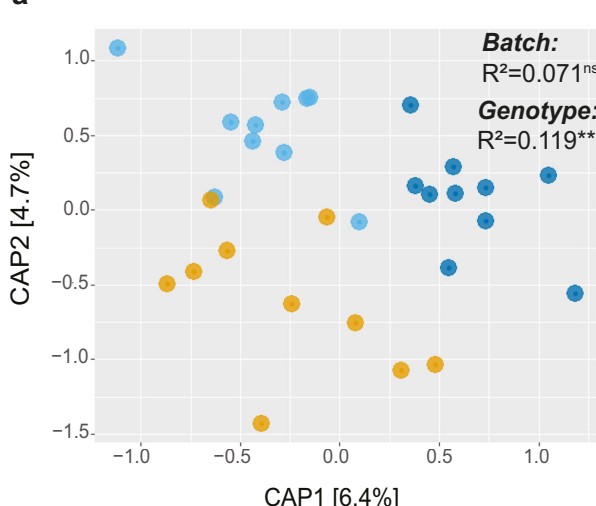

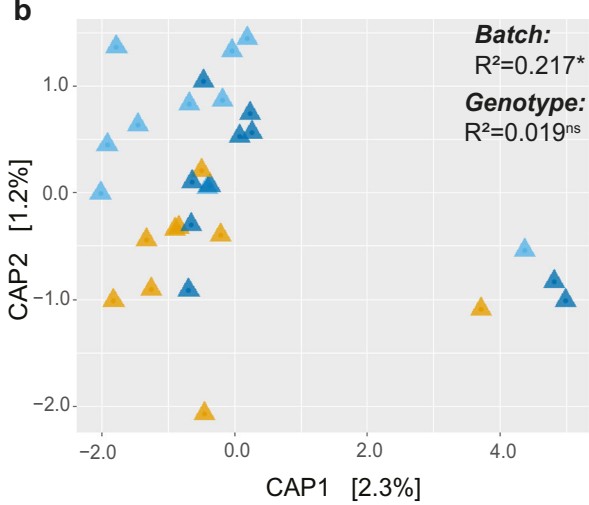

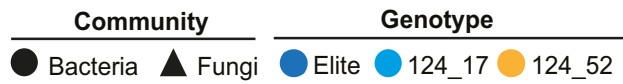

**Fig. 3 Wild alleles at locus *QRMC-3HS* are associated with a shift in the composition of the bacterial, but not fungal, microbiota.** Canonical Analysis of Principal Coordinates computed on Bray–Curtis dissimilarity matrix of **a** bacterial or **b** fungal ASVs' abundances. Sample type is depicted in the bottom part of the figure. The number in the plots show the proportion of variance ($R^2$) explained by the factors 'Batch' and 'Genotype', respectively. Asterisks associated to the $R^2$ value denote its significance, ns not significant. **a** *P*-value 'Batch' = 0.278, F = 1.10; *P*-value 'Genotype' = 0.005, F = 1.84; Adonis test 5000 permutations. **b** *P*-value 'Batch' = 0.027, F = 3.05; *P*-value 'Genotype' = 0.963, F = 0.26; Adonis test 5000 permutations. Source data are provided as a Source data file.

imposed threshold (i.e., Kruskal–Wallis test 2 weeks, $\chi^2 = 12.473$; Kruskal–Wallis test 3 weeks, $\chi^2 = 8.890$ and post hoc Dunn's test, *P*-values < 0.05) (Supplementary Fig. 9a). No significant effect was identified among timepoints, regardless of the type of specimen investigated, i.e., unplanted wash-through or planted exudates (Kruskal–Wallis test, $\chi^2 = 17.761$, *P*-value < 0.05) (Supplementary Fig. 9a, b). Likewise, nitrogen content at the earliest timepoint (2 weeks) ranged from 2.8 to 6.0% (w/w) and was not statistically different between unplanted wash-through or planted samples

(ANOVA, F = 1.035, *P*-value > 0.05) (Supplementary Fig. 9c). We could, however, differentiate among samples at the later timepoint (3 weeks), with a higher nitrogen content of 10–18% (w/w) in the unplanted wash-through, compared to the exudates of planted samples ranging from 4–6% (w/w) compatible with the plant's uptake of this mineral from the nutrient solution. Within these latter specimens, no significant differences among the tested genotypes were identified (Kruskal–Wallis, $\chi^2 = 8.567$, and post hoc Dunn's test, *P*-value < 0.05) (Supplementary Fig. 9d).

We next explored the primary metabolism of the sibling lines and the elite genotype exudates at the onset of stem elongation stage (3 weeks) using gas chromatography–mass spectrometry (GC/MS). The metabolites recovered belong to categories such as amino acids, organic acids, carbohydrates, and other polar compounds (Supplementary Fig. 10). Amongst carbohydrates, fructose and glucose represented the largest fraction of the exudates (Supplementary Fig. 10a). We found the majority of the compounds were classified as amino acids, with L-valine, L-leucine, L-proline, L-isoleucine, L-glutamic and L-aspartic acid as the more abundant (Supplementary Fig. 10b). The main organic acid retrieved was succinic acid (Supplementary Fig. 10c), while gamma-aminobutyric acid (GABA) was the most abundant in the 'other polar compounds' category (Supplementary Fig. 10d). These compounds were present in comparable relative amounts regardless of the genotype, and the genotype effect on the metabolic composition of the exudates was not statistically significant (ANOVA and post hoc Tukey or Kruskal–Wallis and post hoc Dunn's test, *P*-values > 0.05, Supplementary Data 2).

To investigate any potential effect of *QRMC-3HS* on yield, we grew the sibling lines along with the elite cultivar Barke under the same conditions as the microbiota profiling to measure the thousand grain weight (TGW) and main tiller grain weight in four independent experiments (Supplementary Fig. 11). Despite a batch effect identified for one of the replicated experiments, we observed a congruent trend where the elite material had higher yield than the sibling lines, regardless of their allelic composition at *QRMC-3HS* (ANOVA TGW, F = 16.641; ANOVA main tiller grain weight, F = 13.979; post hoc Tukey, *P*-values < 0.05) (Supplementary Fig. 11). We interpret this as an indication that the *QRMC-3HS* alone may not be linked to the yield traits measured. As the sibling lines share a minor proportion (~5%) of wild alleles at other loci, we cannot exclude a contribution of these to the yield phenotype. For instance, we identified an overlap between a yield QTL detected in the same experimental population in the pericentromeric region of chromosome 6H (43.6–52.2 cM)[37] and a region containing a wild introgression in both sibling lines (Supplementary Data 3). Likewise, genes responsible for the seed dispersal attribute of wild barley spikes, designated brittle rachis[38] (HvBtr1–HvBtr2 locus at 40,451,507–40,710,518 bp on chromosome 3H) map physically near to *QRMC-3HS* (33,181,340–36,970,860 bp on cultivar Barke) and may be implicated in the reduced yield observed in the sibling lines. We therefore used molecular markers (see 'Methods') to ascertain haplotype composition for these two genes. This revealed that both sibling lines and the wild parental line carry the wild alleles as they do not show mutations in *Btr1* or *Btr2* (Supplementary Fig. 12). Conversely, our elite parent Barke and four other elite lines used as controls, displayed a mutation in either of these genes, consistent with previous observations (Supplementary Fig. 12).

**Genetic diversity at *QRMC-3HS* is associated with distinct root transcriptional profiles.** To further dissect the genetic mechanisms behind the differences in microbiota recruitment observed between the sibling lines 124_17 (elite-like) and 124_52 (wild-

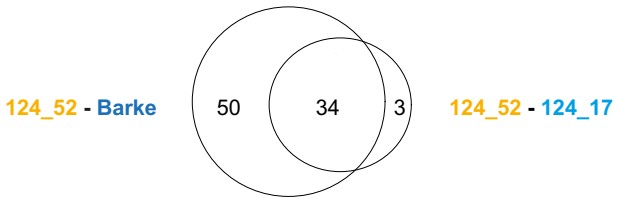

**Fig. 4 The sibling lines harbouring contrasting alleles at locus *QRMC-3HS* and the cultivar Barke display distinct root transcriptional profiles.** Venn diagram showing the number of differentially expressed genes among pairs of comparisons between the sibling lines 124_52 (wild-like), 124_17 (elite-like) and their elite parent Barke (EdgeR pair-wise comparison, individual *P*-values < 0.01, FDR corrected). Source data are provided as a Source data file.

like), we conducted a comparative RNA-seq experiment using root tissue from the sibling lines and Barke. A total of 15 RNA-seq libraries were sequenced, with five biological replicates for each of the three genotypes (Supplementary Table 8). Three comparisons were made: 124_52 vs Barke, 124_17 vs Barke and 124_52 vs 124_17 using the BaRTv2[29] Barke transcriptome as a reference.

Consistent with the high genotypic similarity between the sibling lines and Barke, only 84 BaRTv2 genes were found to be differentially expressed (DE) in the 124_52-Barke comparison, whilst 37 DE genes were identified in the 124_52-124_17 comparison. Interestingly, no DE genes were identified in the 124_17-Barke comparison, and all but three of the DE genes identified in the 124_52-124_17 comparison were also found in the 124_52-Barke comparison (EdgeR, Individual *P*-value < 0.01, FDR corrected, Fig. 4). These results agree with the expectation that lines with the elite *QRMC-3HS* alleles (i.e., the 124_17 sibling line and Barke) have similar transcriptional profiles, with changes in transcription possibly reflecting changes in microbiota compliment (Fig. 3).

A contrasting microbial phenotype was observed between the sibling lines 124_52 and 124_17, despite their similarity at the genetic level (Fig. 3, Supplementary Table 6, Supplementary Data 3). Therefore, we decided to focus on the 34 DE genes shared between the 124_52-124_17 and 124_52-Barke comparisons (Fig. 4), to identify gene products potentially shaping the bacterial microbiota. The full list of 34 DE genes is found in Supplementary Data 4. Of the 34 DE genes in the 124_52-124_17 comparison, only two mapped within the *QRMC-3HS*. The first of these genes is of unknown function, while the second is predicted as a nuclear binding leucine-rich-repeat like (*NLR*).

To understand how underlying genetic differences between 124_52 and 124_17 related to gene expression changes, the allelic composition of these two lines were compared at chromosome 3H (Fig. 5b), and on the other 6 barley chromosomes (Supplementary Figs. 13–18). We mined for regions of contrasting allelic composition in each of the seven chromosomes, and once identified, these were related back to the expression changes of genes expressed in the dataset (Fig. 5, Supplementary Figs. 13–18). We found that the majority (31 of the 34 DE genes) were found on chromosome 3H, and that 28 of these were present at regions of the chromosome with contrasting alleles between 124_52 and 124_17 (Fig. 5). The three DE genes identified on other chromosomes were all found in regions where alleles between 124_52 and 124_17 do not differ. These results suggest that DE between 124_52 and 124_17 is predominantly due to cis-regulation or non-orthologous gene variation (presence/absence variation), and that the number of trans-regulated genes is relatively small. A prediction of this observation is that differences in rhizosphere microbial phenotype between the two

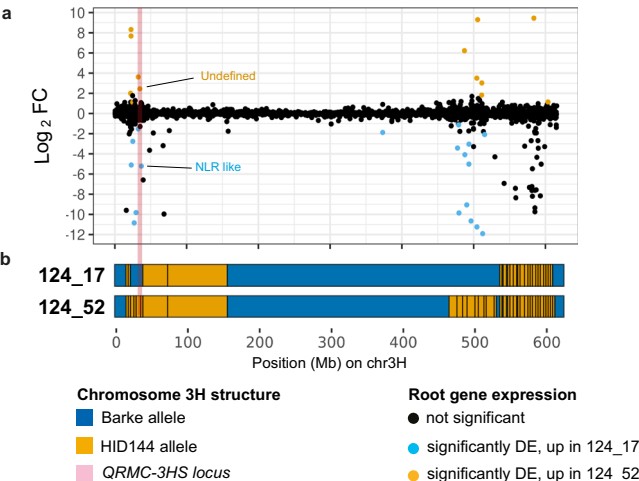

**Fig. 5 Differentially expressed genes mapping at locus *QRMC-3HS*.** **a** Dots depict individual genes and their expression pattern in the pair-wise comparison 124_17 vs. 124_52 (log₂ Fold-Change), colour-coded according to their significance as illustrated at the bottom of the figure (EdgeR, individual *P*-values < 0.01, FDR corrected). **b** Projection of the individual genes on the structures of chromosome 3H for the lines 124_17 (elite-like) and 124_52 (wild-like), respectively, colour-coded according to allelic composition as indicated in the key at the bottom of the figure. The physical location of locus *QRMC-3HS* is highlighted in pale pink. Source data are provided as a Source data file.

lines are not likely due to a large-scale reprogramming of the transcriptome. This is also reflected in the underrepresentation of differentially expressed transcription factors in the 34 124_52-124_17 DE genes (Supplementary Data 4).

**Identification and prioritisation of *QRMC-3HS* candidate genes.** A total of 59 genes were identified in the BaRTv2 gene/transcript[29] annotation within the boundaries of *QRMC-3HS* (identified as 33,181,340–36,970,860 bp on chromosome 3H in the cultivar Barke, Supplementary Data 5). Of these, 25 were found to be expressed in the RNA-seq dataset and were prioritised, as they are likely to be expressed in root tissue (Supplementary Data 5). As previously described, two out of these 25 genes were found to be DE in the 124_52-124_17 comparison subset of 34 genes (Supplementary Data 4).

To further prioritise candidate genes, variant calling was carried out to identify likely high impact and non-synonymous variants between lines 124_52 (wild-like) and 124_17 (elite-like). The variants were annotated with the BaRTv2.18 annotation using SnpEff[39]. A detailed annotation of SNPs identified in each expressed gene is shown in Supplementary Data 6. A total of 545 variants were identified across the 59 BaRTv2 genes annotated within *QRMC-3HS*. However, many of these variants were found in genes not expressed in our RNA-seq data, or were annotated as having low impact, meaning they are either synonymous changes or located in non-coding (5'/3' UTR) regions of genes and were therefore not considered as priority candidates. Two genes carried high-impact variants. BaRT2v18chr3HG123110, of unknown function, has a frameshift variant and a missing stop codon. BaRT2v18chr3HG123140, annotated as a putative Xyloglucan endotransglucosylase/hydrolase enzyme (XTH) carries a frameshift variant close to the 5' end of the coding sequence (CDS) (Supplementary Fig. 19). Fourteen other genes had moderate impact variants, all of which have missense (non-synonymous) SNPs (Supplementary Data 6).

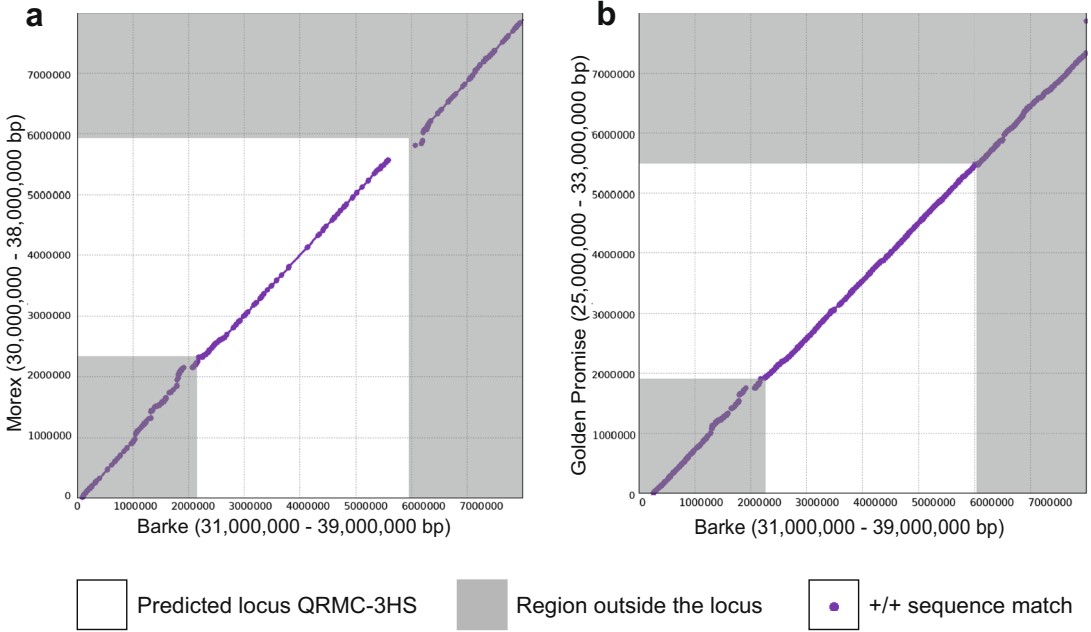

**Fig. 6 Locus *QRMC-3HS* defines an area of structural variation in the barley genome.** Alignment visualisation of the sequence at and surrounding the *QRMC-3HS* locus comparing **a** the cultivars Barke and Morex and **b** Barke to cultivar Golden Promise. The *QRMC-3HS* locus is shown in white, while purple dots represent sequencing matches longer than 1000 bp and ≥95% identity. The gap in the diagonal in (**a**) denotes a disruption of synteny between the two genotypes. Numbers on the axis denote the physical interval, in bp, analysed in the given genomes. Source data are provided as a Source data file.

In summary, three genes in the *QRMC-3HS* were found to either be differentially expressed between two pair-wise comparisons, i.e., 124_52 vs. 124_17 and 124_52 vs. Barke, or have high-impact variants and are therefore considered as primary candidates for shaping the barley rhizosphere microbiota (Supplementary Data 6).

**Structural variation at *QRMC-3HS* in the barley pan-genome.** The recent publication of the barley pan-genome[28] enabled us to investigate potential structural variants at *QRMC-3HS*. These may affect gene presence or expression, and therefore may impact on candidate gene prioritisation. The genome sequence for Barke is represented in the pan-genome, although our wild parent is not. We initially compared the sequence across the *QRMC-3HS* in the cultivar Barke to the corresponding sequence in the cultivar Morex (Fig. 6a). The alignment showed conserved synteny across the *QRMC-3HS* except close to the distal end, where a region of dissimilarity of approximately 480 kb (Barke 3H: 36,582,968–37,063,927 bp) was identified (Fig. 6a). To explore whether this was unique to the Barke-Morex comparison, we compared Barke to 14 other lines in the pan-genome (Supplementary Fig. 20). Comparisons of Barke with Golden Promise (Fig. 6b), Hockett and HOR13942 (Supplementary Fig. 20) showed continuous synteny across *QRMC-3HS*, whilst the other 12 comparisons, including that with the only wild genotype in the pan-genome (B1K-04-12) showed a break in synteny similar to that observed in Morex (Fig. 6a, Supplementary Fig. 20).

The putative *NLR* gene, BaRT2v18chr3HG123500, which was DE between 124_17 (elite-like) and 124_52 (wild-like) (Fig. 5, Supplementary Data 4), has a physical position on chromosome 3H at 36,880,423–36,890,887 bp, within this region of dissimilarity (Supplementary Data 4, 5, Fig. 6a). According to the pan-genome annotation, an ortholog of this gene is not present in Morex, RGT Planet or B1K-04-12. A closer look at the counts per million of the candidate *NLR* revealed that this gene is expressed at low levels in 124_52 (Fig. 7a).

To determine whether this low expression is due to the absence of the gene, we designed a PCR marker specific to a region of the predicted gene BaRT2v18chr3HG123500 (Fig. 7b). We further predicted, based on sequence comparisons (Fig. 6, Supplementary Fig. 20), that the gene would be absent from Morex and RGT Planet, but present in genotypes carrying an elite *QRMC-3HS* (i.e., Barke, Hockett, Golden Promise and 124_17) and so these were included as positive and negative controls, in addition to the wild parental line HID-144. PCR results showed that an amplicon derived from the putative *NLR* gene was not detectable in RGT Planet and Morex, as anticipated from sequence comparisons, while it was present in Barke, Hockett and Golden Promise (Fig. 7c). The amplicon was found to be present in both 124_52 (wild-like) and 124_17 (elite-like) as well as HID-144, albeit with a different size product in HID-144 and 124_52 (Supplementary Fig. 21). This suggests that the difference in *NLR* gene expression between 124_52 and 124_17 may not be due to presence/absence but other polymorphisms in its genomic sequence (Supplementary Fig. 21). Regardless, pan-genome comparisons identify the region at the distal end of the *QRMC-3HS* around the putative *NLR* as a region of sequence divergence.

**Discussion**

In the present study, we combined microbiota abundance and quantitative genetics to identify regions of the barley genome responsible for rhizosphere microbiota recruitment. Our results demonstrate that the taxonomic composition of the rhizosphere microbiota can be treated as a quantitative trait whose genetic basis display structural variants in the barley genome.

Our genetic mapping experiment demonstrated that the heritable component of the barley microbiota in the rhizosphere is controlled by a relatively low number of loci. This is congruent with observations of the bacterial communities inhabiting the phyllosphere of the model plant Arabidopsis[40], the staple crop maize[41] and the cereal sorghum[42]. One of the loci identified in our study, designated *QRMC-3HS*, displays an association with several phylogenetically unrelated bacteria, with the notable

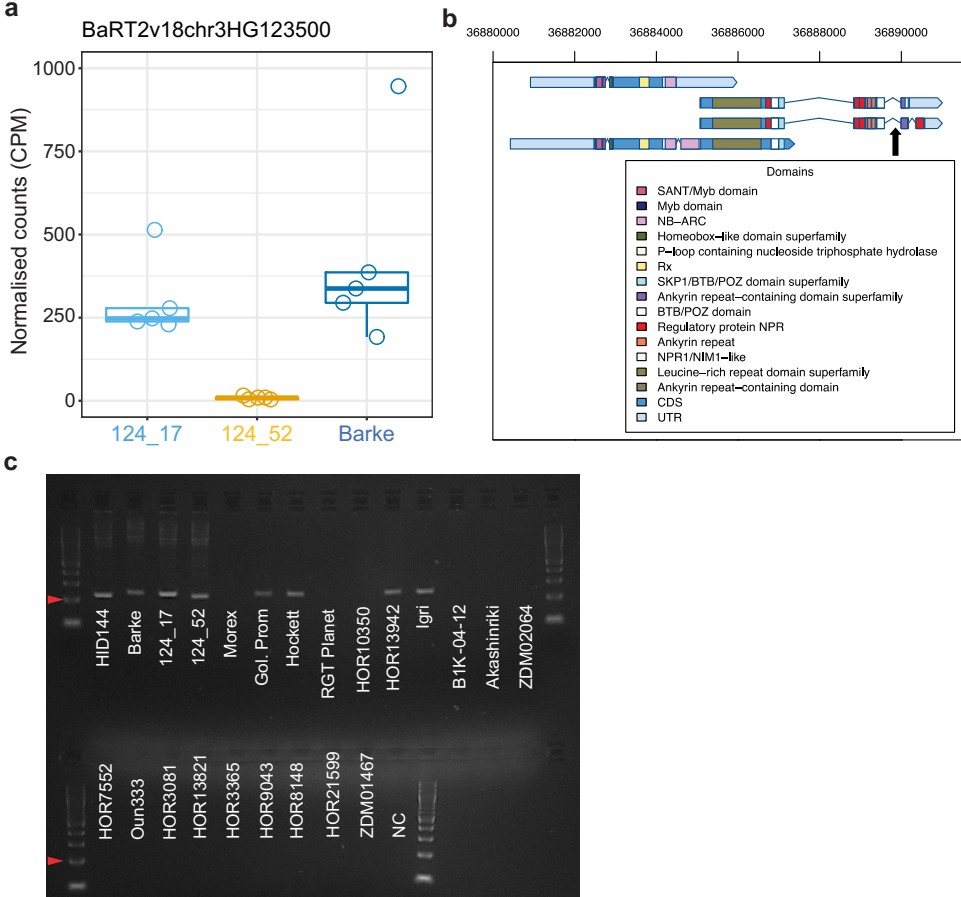

**Fig. 7 The *NLR* gene associated with genotype-dependent transcriptional and genomic variations. a** Boxplot showing the root RNA-seq *NLR* expression across the elite and the sibling lines in normalised counts per million. Individual dots depict individual biological replicates. Upper and lower edges of the box plots represent the upper and lower quartiles, respectively. The bold line within the box denotes the median. Whiskers denote values within 1.5 interquartile ranges. **b** Schematic representation of the *NLR* gene transcripts inferred from the RNA-seq data depicting predicted protein domains from InterProScan. The black arrow depicts the predicted amplicon site of the PCR marker. **c** PCR amplicons partially covering the intron between the LRR and the ankyrin domains in the indicated genotypes-Negative control (NC). A DNA ladder was loaded in the first and last well of each lane, arrowheads indicate the 200 bp fragment. The diagnostic screening was repeated twice with identical results. Source data are provided as a Source data file.

exception of members of Actinobacteria. While the latter are among the bacteria significantly enriched in the elite parent, as previously observed for barley plants grown in the same soil[26], no members of this phylum map at *QRMC-3HS*. A prediction from this observation is that the capacity of soil bacteria to engage with the locus *QRMC-3HS* may be evolutionarily conserved across microbial lineages. This scenario would be congruent with comparative bacterial genomics data which indicates that taxonomically diverse bacteria can share the same adaptive mechanisms to the plant environment[43,44]. An alternative, and not mutually exclusive scenario, is that *QRMC-3HS* mediates the recruitment of a so-called 'microbiota hubs', i.e., individual microorganisms capable of regulating the proliferation of other members of the community, as observed in Arabidopsis[45] and maize[46]. Mining metagenome-assemblies that are significantly associated to plant loci[47] as well as tapping into recently developed synthetic communities of the barley microbiota[48] will enable these scenarios to be investigated experimentally.

The development of powerful genetic and genomic resources allowed us to characterise *QRMC-3HS* at an unprecedented level. We made three important observations. First, the sibling lines harbouring contrasting homozygous alleles at *QRMC-3HS* host contrasting bacterial microbiotas. Despite these lines not triggering the significant enrichment of individual taxa observed in their parental lines, our approach indicates that host genetic composition is sufficient to predict an impact on overall rhizosphere community structure. Besides validating our genetic mapping, this observation is aligned to recent observations of sorghum genotypes[42]. Second, the same lines allowed us to determine that allelic variation at *QRMC-3HS* does not perturb the composition of fungal members of the community. Although this feature is distinct to observations in a genome-wide investigation of root-associated communities of Arabidopsis[49], our finding is consistent with recently reported results for the rhizosphere microbiota of maize, where individual host genes shaped the bacterial but not the fungal microbiota[10]. Third, once we characterised these lines for additional traits that could be intuitively considered to be implicated in shaping the microbiota in the barley rhizosphere, such as root system architecture[30] and rhizodeposition of primary metabolites[50], we failed to identify significant associations between these traits and allelic composition at *QRMC-3HS*. While differences in the genetic background of the tested plants prevent us from drawing firm conclusions when considering these analyses, our observations suggest that *QRMC-3HS* may code for a distinct component of the host genetic control of barley microbiota recruitment.

We therefore employed a root RNA-seq experiment to gain mechanistic insights into the regulation of the microbiota mediated by *QRMC-3HS*. One of the candidate genes found to be significantly up-regulated in plants harbouring elite alleles at *QRMC-3HS* putatively encodes an NLR protein[51]. This class of protein represents one of the two main groups of immune receptors capable of selectively recognising and terminating microbial proliferation via effector recognition[52]. The gene we identified encodes a predicted protein consisting of a Rx-type coiled-coil-nucleotide-binding site-leucine-rich repeat (CC-NLR) domains, containing a putative integrated domain encompassing ankyrin repeats anchoring an NPR1-like (NONEXPRESSOR OF PATHOGENESIS-RELATED GENES 1) domain. This type of integrated *NLR* gene has recently been identified in the wheat genome[53,54]. The integrated domains may work as decoys, mimicking an effector target and enabling microbial recognition[55,56]. The *NPR1* gene is a key transcriptional regulator for plant defence responses related to the hormone salicylic acid (SA)[57]. Besides its canonical role in pathogen protection[58,59], it is important to note that *npr1* mutants, impaired in SA perception, fail to recruit a root microbiota comparable with their cognate wild-type plants[60]. A so-called resistance gene analogue sharing structural features with *bona fide NLR* genes has been identified among the candidate genes underpinning the establishment of the bacterial microbiota in sorghum[42], further suggesting the possible significance of these genes for bacterial recruitment in the rhizosphere. A distinctive feature of the *NLR* gene identified in our study is that it lies in a region of structural variation of the barley genome: for instance, the cultivar Morex, often used as a reference for microbiota investigations[25,26,31] lacks a copy of this gene. The use of the single barley reference genome available prior to 2020[61,62], would have prevented us from identifying this priority candidate gene.

Two other genes within the *QRMC-3HS* were considered among our primary candidates. The first is a gene that is differentially regulated between sibling lines harbouring contrasting microbiotas. As it encodes an unknown protein, we cannot hypothesise its mechanistic contribution to plant-microbe interactions in the rhizosphere. The second is a xyloglucan endo-transglucosylase/hydrolase enzyme (XTH), characterised by a frameshift variant close to the 5' end of the CDS in the wild-like line 124_52. XTH enzymes are widely conserved across plant lineages where they are responsible for cleavage and/or re-arrangement of xyloglucans[63,64], the most abundant hemicellulosic polysaccharides in primary cell walls[65]. In Arabidopsis, cell wall features are a recruitment cue for nearly half of the endogenous root microbiota[66] and cell wall modifications underpin some of the gene ontology categories identified in genome-wide association mapping experiments conducted with this plant[40]. A recent investigation conducted using a so-called 'split-root' approach demonstrated that genes encoding XTH were down-regulated in roots exposed to a high-density microbiome (i.e., akin to natural soil conditions)[67]. Despite not identifying a significant phenotype of the macroscale root system architecture of our sibling lines, whereby XTH may play a primary role, this gene may still contribute to microbiota recruitment via modification of cell wall polysaccharides, a critical checkpoint in molecular plant-microbe interactions[68]. An additional contribution of *XTH* genes to host-microbiota interactions may be represented by an increased adaptation to soil chemical and physical conditions. For instance, *XTH* genes have previously been implicated in abiotic stress tolerance, including drought, salt stress and cold acclimation[69–71]. As wild barley accessions have evolved under marginal soil conditions, these may have imposed a selective pressure on the genetic diversity of *XTH* genes, which, in turn, may have led to a differential microbial recruitment.

As microbiota profiling has not been featuring in breeding programmes, it is legitimate to hypothesize that polymorphisms at candidate genes shaping rhizosphere microbial communities mirror a selection for other, genetically linked, agronomic traits. The observation that *QRMC-3HS* is adjacent to a major QTL for yield-related traits previously identified on chromosome 3H using the same genetic material (*QRMC-3HS*; 38.7–40.6 cM; yield QTL, 40.7–43.9 cM)[72,73] may support this scenario. Selection for yield traits may have inadvertently introduced a gene impacting the microbiota. An alternative, and not mutually exclusive, scenario is that disease resistance may have been the trait under agronomic selection. This would be in line with a recent investigation which demonstrated that bacteria isolated from the barley rhizosphere mediated the establishment of both pathogenic and mutualistic fungi in roots[74]. In this scenario, selection at *QRMC-3HS* may contribute to the fine-tuning of these multitrophic interactions. However, investigations conducted in maize indicate that plant disease resistance is not a reliable predictor of the composition of the phyllosphere microbiota[75] (i.e., the microbial communities populating above-ground plant tissues), suggesting that the activity of individual genetic determinants of the microbiota may be fine-tuned by plant organ-specific mechanisms[76]. Recent innovations in barley genetics[77,78] will facilitate the development of refined genetic material required to probe these scenarios experimentally.

In conclusion, by characterising an experimental population between wild and elite genotypes for rhizosphere microbiota composition, we have identified a putative major plant genetic determinant of the barley microbiota on chromosome 3H. Within the associated interval we have discovered three priority candidate genes, coding for an unknown function protein, an NLR and a XTH enzyme, respectively. These are putatively required for microbiota establishment in wild and cultivated barley genotypes. The latter two proteins have previously been implicated as putative regulators of the microbiota in other plant species. A striking observation derived from our investigation is that one of these candidate genes, the *NLR*, exists in a highly dynamic region of the barley genome, suggesting that selection for agronomic traits may have led to a divergent microbiota in elite cultivars. Our approach can be readily used to identify other or additional candidate genes from reference-quality genomes, including wild ancestors, as they become available for experimentation, in barley and other species. We therefore advocate the use of dedicated plant genetic resources to resolve plant-microbiota interactions at the gene level and accelerate their applications for sustainable crop production.

## Methods

**Plant materials.** Barley plants from family 15 of the nested-association mapping population (NAM) HEB-25[27] were used in this investigation. We developed the sibling lines 124_52 (wild-like) and 124_17 (elite-like), with contrasting haplotypes wild and elite at the *QRMC-3HS*, by selfing the line HEB_15_124 which was heterozygous at the locus of interest. All lines used were genotyped using a combination of KASP markers, Infinium iSelect 9 K and 50 K SNPs arrays platforms. Barley plants used in this study along with the genetic information are summarized in Supplementary Data 3.

**Plants growth conditions and rhizosphere fractionation.** Barley seeds were surface sterilized by serial washings in 70% ethanol (30 s) followed by 5% sodium hypochlorite (15 min). Sterilised seeds were rinsed thoroughly with sterile ddH2O and pre-germinated on Petri dishes containing a semi-solid 0.5% agar solution. Germinated seeds with comparable rootlets were sown in individual 400 mL pots filled with a sieved (15 mm) reference agricultural soil previously used for barley-microbiota investigations and designated Quarryfield[26,30,31]. The number of replicates varies according to the experiment: the mapping experiment $n = 4$ for the parental lines and $n = 2$ for each of the segregating lines, whereas in the sibling lines we used $n = 10$. Unplanted soil pots were included in each experiment and designated bulk soil controls. Plants were grown until stem elongation (~5 weeks, Zadoks 30–35 cereal growth stage) in a glasshouse under the following controlled

environmental conditions: 18/14 °C (day/night) temperature regime with 16 h daylight that was supplemented with artificial lighting to maintain a minimum light intensity of 200 μmol quanta m$^{-2}$ s$^{-1}$. Watering was performed every 2 days with the application of 50 mL of deionized water to each pot. At this developmental stage, plants were uprooted from the soil, stems were detached from the uppermost 6 cm of the root system which, upon removal of large soil aggregates, was subjected to a combination of washing and vortexing to dislodge rhizosphere fractions. Briefly, root material with the adhering rhizosphere was transferred in a sterile 50 mL falcon tube containing 15 mL of phosphate-buffered saline solution (PBS). Samples were then vortexed for 30 s, the soil sedimented for 2–3 min, and the roots transferred in a new 50 mL falcon tube with 15 mL PBS, in which the samples were vortexed again for 30 s to separate the remaining rhizosphere soil from roots. The roots were separated, the two falcon tubes were combined in one single falcon tube, now containing the rhizosphere soil fraction, and then centrifuged at 1500 g for 20 min. After centrifugation, the supernatant was discarded, and the pellet immediately stored at −70 °C. In the unplanted soil controls (i.e., the bulk soil pots), a portion of soil corresponding to the area explored by roots was collected with a spatula and processed as described for planted soils. Total DNA was extracted from the rhizosphere and unplanted soil samples using FastDNA™ SPIN kit for soil (MP Biomedicals, Solon, USA) following the instructions by the manufacturer.

**Assessment of root and yield traits**. Agronomic traits related to yield, brittle rachis and root architecture were assessed for the sibling lines. Main tiller seeds grown in Quarryfield soil ($n = 5$–8, 4 independent replicates) were used to measure yield with a MARVIN seed analyser (Supplementary Fig. 11). Brittle rachis in *Btr1* and *Btr2* gene mutations were assessed using KASP markers designed on *Btr* genomic sequences[79]. Root architecture factors were studied for $n = 4$ plants at early stem elongation for consistency with the microbial rhizosphere experiments (Supplementary Table 7). Roots were thoroughly washed and kept in phosphate-buffered saline solution (PBS) until processing. Root systems were scanned and analysed using WinRHIZO software (Regent Instruments Inc.). Shoot and root dry weights were determined by drying the samples in the oven at 70 °C for 48 h. Specific root length (cm/g) and root density (g/cm³) were calculated computing the ratio of root length and root dry weight and the ratio of dry weight and volume, respectively[18]. Normality was assessed by Shapiro–Wilk test. Significance was tested with a Kruskal–Wallis or an ANOVA test according to data distribution.

**Barley root exudates metabolic profiling**. We developed a protocol to characterise primary metabolites from sand-grown barley plants[80]. Briefly, 3 barley plantlets were sown in a 400 mL plastic pot filled with ~300 g of pre-sterilized silver sand and organised in the glasshouse in randomized blocks design ($n = 15$ per genotype/ block). Barley nutrient solution 100%[81] was applied to each pot (50 mL at 48 h intervals) and in the last week, a 25% nitrogen barley nutrient solution[81] was applied. After 2 and 3 weeks and following the randomized block design, barley roots were carefully taken out of the pots, and the sand around the roots was washed off with water. Using a plastic jar (100 mL vol), root exudates were collected using 3 plants from the same genotype per jar. The washed plant roots were submerged in 50 mL sterilized distilled water and were left to exude for 6 to 7 h. Unplanted controls were generated by washing through the unplanted sand with sterilized distilled water collecting 50 mL, which were processed identically to the exudates. The root exudates (100 mL) and unplanted controls were collected in clean plastic jars, filtered (cellulose Whatman No. 42) and 100 μL of 2 mg/mL erythritol solution (extraction standard for GC/MS) was added to each jar. The exudate solution was frozen at −80 °C and subsequently concentrated to powder form by freeze-drying for 4 days. The experiment was harvested on four consecutive days approximately between 11 AM and 6 PM. Freeze-dried exudates (5 mg) were pooled per genotype and block ($n = 4$) and analysed by an Elemental analyser for total carbon and nitrogen quantification using the Dumas method[82], while 20 mg of the same samples were used to perform a semiquantitative GC/MS analysis as previously described[83]. Briefly, metabolite profiles were acquired using a GC–MS (DSQII Thermo-Finnigan Tempus GC–(TOF)–MS system, UK) system carried on a DB5-MS column (15 m × 0.25 mm × 0.25 μm, J&W, Folsom, CA, USA). Data were acquired using the XCALIBUR (Thermo Scientific, Waltham, MA, USA) software package V. 2.0.7. Semiquantitative data was acquired by integrating selected ion chromatographic peaks.

Data distribution of individual compounds was assessed by Shapiro–Wilk test. Significance was tested with a Kruskal–Wallis with Dunn's test (FDR corrected) or an ANOVA test followed by a Tukey post hoc test according to data distribution.

**Bacterial and fungal DNA quantification**. Bacterial and fungal DNA fractions (total DNA abundance) were quantified in the rhizosphere of the sibling lines by quantitative real-time polymerase chain reaction. Rhizosphere DNA samples were diluted to 10 ng/μL and serial dilutions were applied. A final concentration of 0.1 ng/μL was employed for both the Femto Fungal DNA Quantification Kit and Femto Bacterial DNA Quantification Kit (Zymo Research) according to the manufacturer protocol. The sibling lines DNA samples were randomized in 96-well plates, using a minimum of 10 biological replicates per treatment. Quantification was performed in a StepOne thermocycler (Applied Biosystems by Life

Technology). Data distribution of the DNA samples was assessed by Shapiro–Wilk test. Significance was tested with a Kruskal–Wallis with Dunn's test (FDR corrected).

**Amplicon sequencing library preparation**. 16S rRNA and ITS libraries from rhizosphere and bulk soil preparations were generated using the 515F (GTGCCAGCMGCCGCGGTAA)-806R (GGACTACHVGGGTWTCTAAT) primer pair[84] for amplifying 16S rRNA sequences, while the PCR primers ITS1F (CTTGGTCATTTAGAGGAAGTAA)-ITS2 (GCTGCGTTCTTCATCGATGC) were used for the ITS library[85,86]. Briefly, PCR primer sequences were fused with Illumina flow cell adaptor sequences at their 5′ termini and the 806R primers contained 12-mer unique 'barcode' sequences to enable the multiplexed sequencing of several samples in a single pool.

For each individual bulk and rhizosphere preparation, 50 ng of DNA was subjected to PCR amplification using the Kapa HiFi HotStart PCR kit (Kapa Biosystems, Wilmington, USA). The individual PCR reactions were performed in 20 μL final volume and contained: 4 μL of 5X Kapa HiFi Buffer, 10 μg Bovine Serum Albumin (BSA) (Roche, Mannheim, Germany), 0.6 μL of a 10 mM Kapa dNTPs solution, 0.6 μL of 10 μM solutions of the individual PCR primers, 0.25 μL of Kapa HiFi polymerase. Reactions were performed using the following programme: 94 °C (3 min), followed by 35 cycles of 98 °C (30 s), 50 °C (30 s), 72 °C (1 min) and a final step of 72 °C (10 min). For each primer combination, a no template control (NTC) was subjected to the same process. To minimize potential biases originating during PCR amplifications, individual reactions were performed in triplicate and at least 2 independent sets of triplicate reactions per barcode were performed. To check the amplification and/or any possible contamination, prior to purification, 6 μL aliquots of individual replicates and the corresponding NTCs were inspected on 1.5% agarose gel. Only samples that display the expected amplicon size and no detectable contamination in NTCs on gel were retained for library preparation. Individual PCR amplicons were pooled in a replicate-wise manner and purified using Agencourt AMPure XP Kit (Beckman Coulter, Brea, USA) with 0.7 μL AmPure XP beads per 1 μL of sample. Following purification, 6 μL of each sample was quantified using PicoGreen (Thermo Fisher Scientific, Waltham, USA). Once quantified, individual barcode samples were pooled to a new tube in an equimolar ratio to generate amplicons libraries. Paired-end Illumina sequencing was performed using the Illumina MiSeq system (2 × 150 bp reads) as indicated in ref. [31]. Library pool quality was assessed using a Bioanalyzer (High Sensitivity DNA Chip; Agilent Technologies) and quantified using a Qubit (Thermo Fisher) and qPCR (Kapa Biosystems, Wilmington, USA). Amplicon libraries were spiked with 15% of a 4 pM phiX control solution. The resulting high-quality libraries were run at 10 pM final concentration.

**Amplicon sequencing reads processing**. Quality assessment and DADA2 version 1.10[87] and R 3.5.1[88] was used to generate the ASVs following the basic methodology outlined in the 'DADA2 Pipeline Tutorial (1.10)' and it is explained in detail in ref. [31]. Subsequently, sequences classified as 'Chloroplast' or 'Mitochondria' from the host plant were pruned in silico. Additional pruning was carried out, removing ASVs matching a list of potential contaminants of the lab[89]. Next, we merged the library used for genetic mapping with the library of the sibling lines to perform simultaneous processing of both libraries creating a single new Phyloseq object. Further filtering criterion was applied, low count ASVs were pruned from the merged library (at least 20 reads in 2% of the samples), retaining 93% of the initial reads (Supplementary Fig. 1). This dataset was rarefied at equal sequencing depth across samples (10,500 reads) and agglomerated at genus and family taxonomic levels. Finally, the resulting Phyloseq object was subsetted by the corresponding library for downstream analyses.

The mapping 16S rRNA gene amplicon library merged with the sibling lines library allowed us to identify 2189 individual ASVs from a total of 8,219,883 sequencing reads after filtering and taxonomic identification against the SILVA 138 database[90]. While the sibling lines ITS rRNA amplicon sequencing library was generated identifying 216 individual ASVs from a total of 4,641,285 sequencing reads after filtering and taxonomic identification against the Unite 04.02.2020 database[91].

**Calculation of alpha-, beta-diversity indices and differential abundance between rhizospheres**. Alpha-diversity richness was estimated as described in ref. [31]. Beta-diversity analysis was carried out by calculating the dissimilarities among microbial communities using the rarefied data with the Bray–Curtis index as described in ref. [31]. For ITS, the Bray–Curtis dissimilarity matrix was square-root transformed to allow visualization since all the samples appeared agglomerated in the PCoA visualization. DESeq2 was used to perform microbial differential abundance analysis to identify genera differentially enriched between pair-wise comparisons by Wald test (False Discovery Rate, FDR < 0.05)[92].

**QTL mapping of bacterial microbiota phenotype**. Following the analysis of microbial differential abundance (DESeq2) between wild and elite parental lines, ASVs enriched in the wild or elite parents were recapitulated in the segregant population for further mapping. Association between microbial abundances and loci of the barley genome was conducted using the package R/qtl[93] and the function *scanone*, with the expectation-maximization (EM) algorithm

implementing interval mapping considering only a single-QTL model. The LOD genome-wide significance threshold was set at 20% adjusted per taxa using 1000 permutations. The loci, shown in Fig. 2, were selected based on marker regression analysis or their LOD scores genome adjusted per taxa (Supplementary Tables 1 and 2) (functions *makeqtl*, *fitqtl* and *plotPXG*). The delimitation of the different loci was performed by applying the Plant RNA credible interval method with confidence intervals at 95% (function *bayestint*[94]) (Supplementary Tables 3–5). The percentage of explained variance ($R^2$) was calculated per individual phenotype (taxa), at the flanking maker of the interval upper part, corresponding to each of the individual taxa mapping at this position, which is summarized in Supplementary Tables 3–5.

**Transcriptomic analyses of the sibling lines**. Roots from the cultivar Barke and the sibling lines were processed as described above. Briefly, biological replicates of the different genotypes ($n = 10$) were grown in pots filled with Quarryfield soil and maintained in the glasshouse for 5 weeks in a randomised arrangement. The uppermost 6 cm of the root system were processed as described in 'rhizosphere fractionation'. For harvesting root samples, following vortexing the root system to remove the soil/rhizosphere fractions in PBS, roots were collected with sterile forceps, excess PBS gently removed using a clean paper towel and immediately flash-frozen in liquid nitrogen until processed. All the root systems were collected in 3 consecutive days between 10 AM and 4 PM, reflecting the time necessary to process root samples.

RNA was extracted from individual root systems with the Macherey-Nagel™ NucleoSpin™ RNA, Mini Kit (Thermo Fisher, USA) following the manufacturer's protocol, including the Plant RNA Isolation Aid Invitrogen (Thermo Fisher, USA) for the sample lysis step using 90 μL of Plant RNA Isolation Aid, mixed with 870 μL of RA1 buffer and 40 μL of dithiothreitol (DTT). RNA quality was assessed using an Agilent 2100 Bioanalyzer or TapeStation (Agilent, USA). Samples for sequencing were selected based on microbiota profiles and the quality of the RNA sample.

Approximately, 2 μg of total RNA per sample ($n = 15$) was submitted to Genewiz (Leipzig, Germany) for Illumina sequencing. Total RNA (300 ng) was further purified using the NEBNext mRNA Magnetic Isolation Module (NEB). Library preparation was carried out using the NEBNext® Ultra™ II Directional RNA Library Prep with Sample Purification Beads and indexed with NEBNext Multiplex Oligos for Illumina (96 Unique Dual Index Primer Pairs set 1). Next-Generation sequencing was carried out on an Illumina NovaSeq 6000 using an SP, 300 cycles, flow cell with 2 × 150 bp paired-end reads. The library was stranded with a sequencing depth of 40 M reads per sample.

**Differential expression analysis of RNA-seq**. Downstream data pre-processing and analysis for both transcript quantification and variant calling was carried out using snakemake[95]. The barley reference transcriptome (v2.18) for the cultivar Barke was obtained from https://ics.hutton.ac.uk/barleyrtd/bart_v2_18.html. Raw reads were trimmed using trim galore (https://github.com/FelixKrueger/TrimGalore) version 0.6.6 with parameters "-q 20 –Illumina –paired". Transcript quantification was carried out using Salmon[96] version 1.4.0 using parameters "-l A –seqBias –posBias –validateMappings" with BaRTv2.18[29] as the reference transcriptome. Analysis of RNA-seq quantifications was carried out using a custom modified version of the 3D RNAseq pipeline[97]. The tximport R package version 1.10.0 was used to import transcript TPM values and generate gene TPM values[98]. Low expressed transcripts and genes were filtered based on analysing the data mean-variance trend. The expected decreasing trend between data mean and variance was observed when transcripts which had <3 of the 15 samples with counts per million reads of 2 were removed, which provided an optimal filter of low expression. A gene was counted as expressed and included in the down-stream differential expression (DE) analysis if any of its transcripts met the above criteria. The TMM method was used to normalise the gene and transcript read counts to -CPM[99]. The R package umap (https://cran.r-project.org/web/packages/umap/vignettes/umap.html) implementing the umap algorithm[100] was used to visualise the expression data. It was found that sample date influenced gene expression and so this was incorporated into the EdgeR linear model as a block effect.

DE analysis was carried out using the R package EdgeR[101] version 3.32.0. The EdgeR generalised linear model quasi-likelihood (glmQL) method was used, with genotype and date of sampling used as terms in the model (~0 + genotype + sampling.date). Contrast groups were set to 124_17-Barke, 124_52-Barke and 124_52-124_17. P-values were corrected using the Benjamini–Hochberg method to correct the false discovery rate[102]. Genes were considered to be DE if they had an adjusted P-value < 0.01 and a Log2FC > =1 or < =−1 (Figs. 4, 5 and Supplementary Figs. 13–18).

**Variant calling**. For variant calling the trimmed Illumina reads were combined according to genotype (Barke, 124_17 or 124_52) using the Linux cat command (forward and reverse reads in separate files). Mapping was carried out against the barley Barke genome[28] with STAR version 2.7.9[103]. To aid with read mapping, a BaRTv2.18 gtf file was used with the "genomeGenerate" mode. After an initial round of mapping was carried out, splice junctions from each of the genotypes were collated and filtered using a custom script, removing splice junctions with

non-canonical dinucleotide sequences, those with a read depth <4 and a max overhang <10 bp. The filtered splice junction set was used as input for a second round of mapping.

Mapped read files (.bam files) were pre-processed prior to variant calling using Opossum[104] with settings "SoftClipsExist True". Variant calling was carried out using Platypus[105] with the barley Barke genome as a reference, and with settings "–filterDuplicates 0 –minMapQual 0 –minFlank 0 –maxReadLength 500 –minGoodQualBases 10 –minBaseQual 20". Variant calling files (VCF) were merged using bcftools merge and filtered to remove variants outside the QRMC-3HS locus (Barke chr3H: 33,181,340–36,970,860 bp). The resulting QRMC-3HS VCF was filtered to only keep variants above the quality threshold, and where genotypes Barke and 124_17 (elite-like) were called as reference alleles and 124_52 (wild-like) was called as the alternative alleles. InterProScan version 5.48-83.0 (version 83.0 data) was used to predict functional domains of predicted proteins from transcripts.

**MUMmer alignment**. To ascertain the physical position of QRMC-3HS in each genome, the sequences of two markers flanking the locus (i.e., SCRI_RS_141171 and SCRI_RS_154747) were aligned to reference genomes from the pan-genome[28] using BLAST[28,106] with default parameters. The best alignment for each flanking marker was selected as the physical position in each case. For the purposes of visualisation, these numbers were rounded to the nearest Mb. The sequence +/− 2 Mb on either side of the flanking markers was extracted from each genome using a custom python script. The programme NUCmer from the MUMmer suite[107] was used to align the QRMC-3HS sequence from each of the pan-genome lines against the QRMC-3HS sequence of Barke with settings "dnadiff". The resulting delta file was filtered using delta-filter with the settings "-I 95 -l 1000 -g", resulting in all alignments with <95% identity and lengths of <1000 bp being removed. The programme mummerplot was then used to create figures (Fig. 6 and Supplementary Fig. 20).

**NLR diagnostic marker**. The NLR candidate gene diagnostic PCR marker was designed to amplify the flanking region of an 18 nt deletion located in the fourth intron of the predicted gene in line 124_52 compared with the elite parent Barke (Supplementary Fig. 21). Seedlings of the barley genotypes of the pan-genome collection[28] were grown under controlled conditions and young leaves subjected to DNA extraction using the DNeasy Qiagen Plant Kit. The primers designated 'forward' (GCCTTTTCAGCAAGATGCCG) and 'reverse' (GTACTCCCTCCGCTCCAAAAT) were used to perform PCR amplifications with the Kapa HiFi HotStart PCR kit (Kapa Biosystems, Roche). The reactions were performed in a SimpliAmp Thermal Cycler (Applied Biosystems) using the following conditions: 94 °C (3 min), followed by 30 cycles of 98 °C (30 s) denaturing, 65 °C (30 s) annealing, 72 °C (30 s) elongation and a final elongation step of 72 °C (10 min). PCR amplicons were separated and visualised in a 2% agarose gel (Fig. 7c).

**Reporting summary**. Further information on research design is available in the Nature Research Reporting Summary linked to this article.

## Data availability
The raw sequence data collected in this study have been deposited in the European Nucleotide Archive (ENA) accession number PRJEB50061. The barley reference transcriptome (v2.18) for the cultivar Barke was obtained from Barley Reference Transcript (BaRTv2.18) Dataset [https://ics.hutton.ac.uk/barleyrtd/bart_v2_18.html]. Pseudomolecules of individual barley genomes were downloaded from https://webblast.ipk-gatersleben.de/downloads/barley_pangenome/. Source data are provided with this paper.

## Code availability
The codes to reproduce the figures and statistical analyses are available in the GitHub repository [https://github.com/BulgarelliD-Lab/Microbiota_mapping][108].

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

## Acknowledgements

We thank Malcolm Macaulay and Jim Wilde (The James Hutton Institute, Invergowrie, UK) for the technical assistance during the execution of the experiments. We also thank Nils Stein and Mary Ziems (IPK, Gatersleben, Germany) who kindly provided us with seeds of the barley pan-genome collection. In addition, we thank Malcolm Macaulay and Luke Ramsay (The James Hutton Institute, Invergowrie, UK) who provided access to the diagnostic technology for Btr genes. This work was supported by a Royal Society of Edinburgh Personal Research Fellowship to D.B. and an UKRI grant (BB/S002871/1) to D.B., G.B. and R.W.

## Author contributions

C.E.M., R.A.T., R.W. and D.B. designed the experiments. C.E.M., M.C., R.S., J.A., G.B. and D.B. conceived the data analyses. C.E.M., R.A.T., R.K., L.P., M.M., A.A., performed the experiments. C.E.M., M.C., A.F., R.K., A.A., R.S., G.N, T.M. and J.A. analysed the data. J.M. and P.H. generated the amplicon sequencing libraries. A.M and K.P. provided access to the HEB-25 seed material and analysed their genomic information. C.E.M., M.C. and D.B. wrote the first version of the manuscript, all authors collegially commented on and contributed to the submitted version.

## Competing interests

The authors declare no competing interests.
