## [Peer Review File · Nature Communications]

IDENTIFYING PLANT GENES SHAPING MICROBIOTA COMPOSITION IN THE BARLEY RHIZOSPHEREReviewers' Comments:

Reviewer #1:

Remarks to the Author:

The paper by Escudero-Martinez et al describes the use of classic QTL mapping approaches to identify and partially characterize candidate loci underlying rhizosphere microbiome composition in barley. The parent lines of the mapping population were an elite domesticated cultivar and a wild barley relative (Barke and HID144 respectively). Special attention is paid to the most promising QTL, designated QRMC-3HS, which was localized to a structural variant that segregates among domesticated cultivars of barley. Near-isogenic lines segregating for QRMC-3HS were created and shown to differ in bacterial microbiome composition.

This is a thorough, high-quality, and very interesting study that advances our knowledge of host plant genes that influence microbiome assembly (and perhaps microbiome function, although that was not investigated here). I enjoyed reading it. The methodology is largely robust and well described (a couple exceptions noted below), including the statistical analyses which are appropriate. A logical progression of experiments validate the association of the QRMC-3HS locus with microbiome composition, rule out certain physiological mechanisms (e.g. root exudates, root architecture), and identify three possible candidate genes within the locus, one of which is a NLR. This was surprising to me because I am familiar with the activity of NLRs in effector-triggered immunity inside the plant, yet the rhizosphere community is clearly external to the plant; but the Discussion helped to clarify that this paper adds to the growing evidence that NLRs may have additional microbiome-related function in the rhizosphere. Overall I'm enthusiastic about this work, but have a few suggestions for improvement:

My main concern about this manuscript is related to the inference that variation at QRMC-3HS affects the abundance of "several phylogenetically unrelated bacteria". Although this inference is plausible, because the phenotype here is measured using sequencing data, the measured abundances of separate ASVs/taxa are actually non-independent from each other: for a given number of sequence reads (a ceiling set by the output of the sequencer, unrelated to the biology of the system), if the true abundance of one taxon goes up, then the measured abundances of all other taxa will go down. This is especially a problem for abundant taxa and it is unclear how abundant were the taxa associated with QRMC-3HS. This well-known compositionality problem creates spurious negative correlations between features, which in turn could create the illusion of a shared genetic basis where there is none. If the counts were to be corrected for their compositional nature (see e.g., DOI: [10.1093/gigascience/giz107](https://doi.org/10.1093/gigascience/giz107)) would this conclusion still hold, or would it become clearer that QRMC-3HS affects only a particular group of bacteria?

Related, the authors note that QRMC-3HS also associates with higher taxonomic levels above ASVs - e.g., family (line 135-144). The implications of this are not well explained, what does this tell us about the possible mechanisms underlying this locus? This question is especially interesting given that one candidate gene in the locus is a NLR, which usually interact only with a narrow taxonomic range of microbes, often only a single strain.

In the Introduction and Discussion, the inferences about the role of "human selection" are not really justified. For one thing, is HID144 the progenitor of modern barley or just a wild relative? The fact that the structural variant containing QRMC-3HS is still segregating among modern barley lines shows that this locus was not fixed during domestication. It is also notable that QRMC-3HS is closely linked to a known yield locus. In my opinion there is not yet good evidence that QRMC-3HS has been under selection - for example there is no population-genomic analysis or test of the effects on microbiome function.

Minor comments:

Line 160, 181, 189 The term "16S rRNA gene copy numbers" is a bit ambiguous, as "copy number" usually refers to CNV among bacterial species. Until reading the methods it is not clear this was meant to estimate the total microbial abundance. Also note, this metric will also be influenced by variation in the 16S gene CNV among bacterial species, so it could also reflect differences in community composition. The same is true for ITS "copy number"

Lines 254-262 this result could also be explained by the remaining ~5% of wild barley genome that is retained in the near-isogenic lines - not just the region flanking QRMC-3HS

Line 279 and elsewhere: To help the reader interpret the results, it would be good to name the two near-isogenic sibling lines according to which allele they have at QRMC-3HS (rather than just 52 and 17).

Lines 328-329: does this potentially create false negatives, due to the exclusion of cis-regulatory sites in the 5' UTRs?

Line 387: What was the total heritability of the microbiome components associated with QRMC-3HS? Is there missing heritability? It seems likely that the heritable part of the rhizosphere microbiome would also be affected by numerous loci with effects that are too small to be detected in a study like this (the same is true for many quantitative traits)

Lines 391-394: I find this difficult to understand - does this mean that Actinobacteria were not associated with QRMC-3HS, but in other studies they did differ between wild/domesticated lines? I recommend re-wording this for clarity

Lines 481-483: but there is a notable counterexample too, where the maize phyllosphere microbiome was mostly robust to introgressed disease-resistance loci: <https://doi.org/10.1111/nph.16284>

Reviewer #2:

Remarks to the Author:

the authors investigated the microbiota of wild and domesticated barley, using metagenomics as an external phenotype. They identified a concise number of ASVs different between the parent lines. A segregating population between the two parents was generated, and their microbiome composition was correlated with genomic differences. One QTL region was identified that was robustly found at various taxonomic levels. To understand the function of the QTL, the authors generated sibling lines from a line heterogenous for this locus. The lines were similar in root morphology and exudation, but differed in microbiome composition and transcriptional profile. Again, only a concise number of genes was distinct between the lines as well as the sibling lines and the elite cultivar. Mapping of the differentially expressed genes pointed towards three genes found in the QTL region, one of them being a NLR gene likely involved in microbe perception and immune signaling. This candidate is close to a region important for yield. Selection for high yield in agriculture might thus have resulted in a distinct microbiota in barley cultivars.

The manuscript is well written, the figures well designed, and the conclusions well supported by data. The authors combine a number of recent technologies to identify host genes involved in shaping the microbiome. The approaches presented could be transferred to other plant species easily. The biological findings as well as the methods applied will be of interest to a broad readership.

Detailed comments:

Fig 1a: it seems that the microbiota are among others dominated by the large, gray ASV situated between elite and wild parents. what is that?

Line 108, 1b PCA: the segregant population seems to cluster around the elite parent, with quite some

distance to the wild parent. Do you have an explanation for this?

supp table 4: please indicate number of roots analyzed.

supp fig 4a: I am no expert at all on QTL analysis and thus my question likely has an obvious answer: isn't the heterozygous allele supposed to be a mix between elite and wild parents, and thus to be expected to localize at an intermediate level on the plots?

L236: please elaborate on the negative (soil) controls of this experiment. were metabolites not detected at all in pots without plants, or were levels significantly lower?

L236: I assume in the GCMS experiment, some 100-1000 features were identified. were differences between the lines found when all features are taken into consideration?

L455: typo: hypothesize

regards,
Joelle Schläpfer

Reviewer #3:

Remarks to the Author:

Plant and its surrounding microorganisms form complex networks which is important for plant adaptation to the environments. It is well known that different plants have distinguish microbiota and different crop production systems also have fundamental impact on the microbiota in the rhizosphere. It is much unknown how plant genes shape the microbiota, which is critical for genetic improvement of crops for environmental adaptation. This paper first used metagenomics information as an external quantitative phenotype to map the genetic loci to determine microbiota profiles and then identify the quantitative loci and candidate genes. It would be ideal if the authors can confirm some of the genes' functions. However, this is not essential for the paper to be accepted for publication. The experimental designs and data analysis are robust and the results are well supported by the experimental data. I have only a few minor suggestions for the authors.

1. The result for QTL mapping is too simple. The authors should provide more information for Fig 2. The current presentation did not reflect the complexity of plant and microbiota interaction.

2. The authors over emphasized NLR gene and biotic stress tolerance in the discussion and other parts of the paper. I suggest that the authors consider a balance presentation of the plant genes for both biotic and abiotic stress tolerances and adaptation. The abiotic stress and plant interaction may play more important role to shape the microbiota in rhizosphere. There are some reports that xyloglucan endotransglucosylase/hydrolase enzyme is involved in abiotic stress responses, e.g. drought tolerance. Thus, the discussion part need expand the discussion on this gene.

3. The pan genome analysis is good to identify some structural variation in the QTL region, but without the genome sequence of the wild barley donor, there is limitation for the analysis, e.g. missing some key genes in the QTL region based on the other genomes sequences.

4. There are some minor spelling or grammar errors, e.g. allele vs alleles

Point-to-point response letter for the manuscript 'Identifying Plant Genes Shaping Microbiota Composition In The Barley Rhizosphere'

First and foremost, we would like to thank the Editor and three Reviewers for their time and the insightful and constructive comments on our submission. Here, we provide our answers to and interpretation of the individual questions and concerns raised. The Reviewer's comments have been italicized while authors' responses are non-italicized. When additional references were included to support our responses, we have provided individual PubMed identifiers. Unless otherwise specified, line numbering in the authors' responses refers to lines in the revised document with 'track-changes'.

Reviewer #1

The paper by Escudero-Martinez et al describes the use of classic QTL mapping approaches to identify and partially characterize candidate loci underlying rhizosphere microbiome composition in barley. The parent lines of the mapping population were an elite domesticated cultivar and a wild barley relative (Barke and HID144 respectively). Special attention is paid to the most promising QTL, designated QRMC-3HS, which was localized to a structural variant that segregates among domesticated cultivars of barley. Near-isogenic lines segregating for QRMC-3HS were created and shown to differ in bacterial microbiome composition.

This is a thorough, high-quality, and very interesting study that advances our knowledge of host plant genes that influence microbiome assembly (and perhaps microbiome function, although that was not investigated here). I enjoyed reading it. The methodology is largely robust and well described (a couple exceptions noted below), including the statistical analyses which are appropriate. A logical progression of experiments validate the association of the QRMC-3HS locus with microbiome composition, rule out certain physiological mechanisms (e.g. root exudates, root architecture), and identify three possible candidate genes within the locus, one of which is a NLR. This was surprising to me because I am familiar with the activity of NLRs in effector-triggered immunity inside the plant, yet the rhizosphere community is clearly external to the plant; but the Discussion helped to clarify that this paper adds to the growing evidence

that NLRs may have additional microbiome-related function in the rhizosphere. Overall I'm enthusiastic about this work, but have a few suggestions for improvement:

Many thanks for the accurate summary and the very positive evaluation! We acted on the comments/suggestions as indicated in the individual points below.

My main concern about this manuscript is related to the inference that variation at QRMC-3HS affects the abundance of "several phylogenetically unrelated bacteria". Although this inference is plausible, because the phenotype here is measured using sequencing data, the measured abundances of separate ASVs/taxa are actually non-independent from each other: for a given number of sequence reads (a ceiling set by the output of the sequencer, unrelated to the biology of the system), if the true abundance of one taxon goes up, then the measured abundances of all other taxa will go down. This is especially a problem for abundant taxa and it is unclear how abundant were the taxa associated with QRMC-3HS. This well-known compositionality problem creates spurious negative correlations between features, which in turn could create the illusion of a shared genetic basis where there is none. If the counts were to be corrected for their compositional nature (see e.g., DOI:10.1093/gigascience/giz107) would this conclusion still hold, or would it become clearer that QRMC-3HS affects only a particular group of bacteria?

Many thanks for having raised this important point. As stated in the original submission:

L133-36: 'We observed up to four unrelated ASVs representing 5.68% of the enriched bacterial reads in the parental lines were linked to the QRMC-3HS locus...'

Thus, it is fair to assume that those taxa do not represent overly dominant members of the microbiota. Having clarified this point, we agree with the Reviewer that microbiota data are inherently compositional. However, the benefits and trade-off of including compositionality in the statistical analysis of microbiota data are still a matter of debate: while studies illustrated that compositional differential abundance methods may outperform the accuracy of non-compositional (e.g., PMID:35039521), other investigations did not support this conclusion (e.g., PMID:32746888). This is due, at least in part, to the experimental design investigated. For instance, simulated and real data indicate that the tool we used, *i.e.*, DESeq2, provides increased sensitivity on a relatively small dataset, *i.e.*, less than 10 replicates/sample, a

common instance in our and many other plant microbiota investigations (e.g., PMID:31657460) when compared to other approaches, including compositional ones (e.g., PMID:28253908).

To gain empirical insights into this aspect of the investigation, we have conducted a new *in silico* experiment and performed a differential abundance analysis between the parental lines (*i.e.*, HID-144 and Barke) of our mapping population using ALDEx2 as suggested by the reviewer and proposed in PMID:31544212. This type of analysis considers our data as compositional and the results are shown in the Venn diagrams presented in Additional Figure 1 below. In summary, when a compositional differential abundance method was applied, we obtained a similar number of taxa as being differentially abundant (DA), however none of those DA taxa were retained when an FDR<0.3 was applied to the ALDEx2 output. Conversely, the DESeq2 was corrected for FDR <0.05.

Additional Figure 1: Venn diagrams illustrating the number of differentially abundant ASVs significantly enriched in either Barke or HID-144 retrieved using DESeq2 and ALDEx2 analyses, respectively. The analyses in ALDEx2 were performed with three different methods for the centred log-ratio (clr) transformation: the geometric mean abundance of all features (“all”), median abundance of all features (“median”) and the features in the inter-quartile of the variance of the clr values across all samples (“iqlr”), respectively. The threshold for the individual *P*-values of differentially abundant ASVs was set to <0.05 (DESeq2, FDR corrected; ALDEx2 no multiple correction applied).

We observed that less abundant ASVs were particularly impacted by this approach, including the ones mapping at the *QRMC-3HS*, compared with DESeq2. Whereas more abundant taxa were more conserved between analyses. These observations are aligned with data from the literature indicating the conservative nature of compositional tools for data analysis (e.g., PMID:35039521). As the plant genotype has a limited impact on the composition of the rhizosphere microbiota (i.e., ~5-15%, e.g., PMID:34287929; PMID:25974302), the application of conservative methods may therefore fail to recapitulate genuine biological diversity. As the locus identified with DESeq2 analysis was further validated with the sibling lines, we argue that for this type of investigation a non-compositional tool may represent the more accurate choice. We however refrain from inferring general principles: we cannot exclude that with an extended number of replicates, different soil (and beyond barley, different host species) compositional methods may outperform DESeq2.

Related, the authors note that QRMC-3HS also associates with higher taxonomic levels above ASVs - e.g., family (line 135-144). The implications of this are not well explained, what does this tell us about the possible mechanisms underlying this locus? This question is especially interesting given that one candidate gene in the locus is a NLR, which usually interact only with a narrow taxonomic range of microbes, often only a single strain.

We have now further clarified the original section of the discussion where we elaborated on the implications of higher taxonomic ranks of the microbiota mapping at *QRMC-3HS* as follow:

L414-22 'A prediction from this observation is that the capacity of soil bacteria to engage with the locus *QRMC-3HS* may be evolutionarily conserved across microbial lineages. This scenario would be congruent with comparative bacterial genomics data which indicates that taxonomically diverse bacteria can share the same adaptive mechanisms to the plant environment. An alternative, and not mutually exclusive scenario, is that *QRMC-3HS* mediates the recruitment of a so-called "microbiota hub", i.e., individual microorganisms capable of regulating the proliferation of other members of the community, as observed in *Arabidopsis* and maize'.

This latter scenario (i.e., locus *QRMC-3HS* controlling 'hubs' of the microbiota) would provide a mechanistic explanation to the relatively 'diverse taxonomic specificity' of the *NLR* candidate.

In addition, the identified candidate *NLR* gene is predicted to be characterised by having an integrated domain *NPR1*-like. This integrated domain may provide further mechanistic insights into the taxonomic specificity of the *NLR* candidate as *NPR1* is itself a target for bacterial effectors (e.g., PMID:29174403) and independently evolved effectors converge in host immunity targets (e.g., PMID:3170753). Finally, we cannot exclude that the role played by *NPR1*, and hypothetically by an integrated domain sharing similarity with said gene, in the metabolism of Salicylic Acid, this latter a determinant of the plant microbiota, as demonstrated in other plant models (e.g., PMID: 26184915). We discussed these concepts in the original submission and those are retained on L454-64 of the revised manuscript.

In the Introduction and Discussion, the inferences about the role of “human selection” are not really justified. For one thing, is HID144 the progenitor of modern barley or just a wild relative? The fact that the structural variant containing QRMC-3HS is still segregating among modern barley lines shows that this locus was not fixed during domestication. It is also notable that QRMC-3HS is closely linked to a known yield locus. In my opinion there is not yet good evidence that QRMC-3HS has been under selection - for example there is no population-genomic analysis or test of the effects on microbiome function.

The accession HID-144 belongs to the subspecies *Hordeum vulgare* ssp. *spontaneum*, the progenitor of domesticated barley (i.e., *H. vulgare* ssp. *vulgare*). We have now rectified this lack of information by adding species names in lines 72-73.

We concur with the referee reasoning that, in the absence of additional experimentation, it is premature to draw the correlation between structural variation at *QRMC-3HS* and domestication/breeding selection. As we consider said additional experimentation beyond the scope of this submission, we rectified instances of ‘human selection’ as follows:

L399-402: ‘Our results demonstrate that the taxonomic composition of the rhizosphere microbiota can be treated as a quantitative trait whose genetic basis display structural variants in the barley genome’.

L501-04: 'As microbiota profiling has not been featuring in breeding programmes, it is legitimate to hypothesize that polymorphisms at candidate genes shaping rhizosphere microbial communities mirror a selection for other, genetically linked, agronomic traits'.

Line 160, 181, 189 The term "16S rRNA gene copy numbers" is a bit ambiguous, as "copy number" usually refers to CNV among bacterial species. Until reading the methods it is not clear this was meant to estimate the total microbial abundance. Also note, this metric will also be influenced by variation in the 16S gene CNV among bacterial species, so it could also reflect differences in community composition. The same is true for ITS "copy number"

We have changed "copy number" by bacterial/fungal total abundance throughout the text as it better reflects the experimental procedure applied. While we acknowledge the 16S rRNA genes and ITS sequences copy numbers among individual taxa can vary, this remains a reference approach to infer the quantity of microbial DNA for comparative purposes (e.g., PMID: 32252812) which was the rationale of this experimentation.

Lines 254-262 this result could also be explained by the remaining ~5% of wild barley genome that is retained in the near-isogenic lines - not just the region flanking QRMC-3HS

This is an excellent observation! We performed an experiment *in silico* and, alongside the introgression encompassing the *Btr* locus described in the original submission, we identified a yield QTL mapping in the same experimental population to the pericentromeric region of chromosome 6H, (PMID:29767798; 43.6-52.2 cM). Although not associated with a differential bacterial enrichment, this region corresponds to a wild introgression in both sibling lines (Supplementary Database 1, current submission). We have therefore revised this sentence as:

L267-72: 'As the sibling lines share a minor proportion (~5%) of wild alleles at other loci, we cannot exclude a contribution of these to the yield phenotype. For instance, we identified an overlap between a yield QTL detected in the same experimental population in the pericentromeric region of chromosome 6H (43.6-52.2 cM) and a region containing a wild introgression in both sibling lines (Supplementary Database 1).'

Line 279 and elsewhere: To help the reader interpret the results, it would be good to name the two near-isogenic sibling lines according to which allele they have at QRMC-3HS (rather than just 52 and 17).

Thank you for the suggestion: we have included wild-like and elite-like next to the sibling lines names in the first occurrence of each paragraph and figure captions.

Lines 328-329: does this potentially create false negatives, due to the exclusion of cis-regulatory sites in the 5' UTRs?

We disagree with this interpretation. In fact, we did not exclude variants at putative cis-regulatory sites as those were listed in Supplementary Database 5 of the original submission and retained in the revised version. Furthermore, if these 5' UTRs variants impacted a cis-regulatory site, genes modulated by those polymorphic cis-regulatory elements should have displayed differential expression, and, as such, been retained in the analysis. For this reason, we are confident we haven't excluded primary candidates from the investigation by not prioritising variants in putative 5' UTRs.

Line 387: What was the total heritability of the microbiome components associated with QRMC-3HS? Is there missing heritability? It seems likely that the heritable part of the rhizosphere microbiome would also be affected by numerous loci with effects that are too small to be detected in a study like this (the same is true for many quantitative traits)

We operationally defined the heritability of each taxon as the proportion of phenotypic variance (*i.e.*, sequencing reads) attributable to individual QTLs. For each taxon, we obtained this value, designated R^2 , from a linear regression model (*e.g.*, PMID: 29767798). These values were tabulated and included, alongside other QTLs information, in Supplementary Table 2 of the original submission. As only a fraction of microbiota composition, *i.e.*, ~5% of sequencing reads, map at QRMC-3HS we consider it more accurate to provide individual R^2 rather than their cumulative values. We have now added an extra sentence in the results section explaining this concept and pointing the reader to the individual data:

L145-47 ‘...and explaining at least 20% of the phenotypic variance (*i.e.*, sequencing reads) for the individual taxa significantly associated to it (Supplementary Table 2)’.

Regarding the concept of missing heritability, this was interpreted as phenotypic variance explained by genetics other than the one of the parental lines (*e.g.*, PMID:20479774). We recognise this point is gaining centre stage in quantitative genetics: however, it cannot be tested as introgression lines harbouring contrasting alleles (other than the ones of the parental lines) are not available for experimentation.

Lines 391-94: I find this difficult to understand - does this mean that Actinobacteria were not associated with QRMC-3HS, but in other studies they did differ between wild/domesticated lines? I recommend re-wording this for clarity

Yes, this is indeed the scenario summarised by the reviewer and we modified the sentence as following:

L407-11 ‘One of the loci identified in our study, designated *QRMC-3HS*, displays an association with several phylogenetically unrelated bacteria, with the notable exception of members of Actinobacteria. While the latter are among the bacteria significantly enriched in the elite parent, as previously observed for barley plants grown in the same soil, no members of this phylum map at *QRMC-3HS*’.

Lines 481-83: but there is a notable counterexample too, where the maize phyllosphere microbiome was mostly robust to introgressed disease-resistance loci:
<https://doi.org/10.1111/nph.16284>

We thank the reviewer for having suggested this additional reference which is indeed pertinent to the discussion. However, the fact that neither gene(s) underpinning the MDR locus described in the manuscript suggested by the reviewer, nor gene(s) underpinning our locus have been cloned, prevents us discussing on first principles. However, it is important to mention that components of the plant disease resistance machinery are influenced by organ-specific mechanisms. For instance, members of the botanical family Poaceae, to which barley belongs, preferentially express *NLR* genes in roots (PMID: 29187571). Thus, in the absence of further

genetic details on the two loci, we believe that the most parsimonious interpretation of the observation suggested by the reviewer is the following one, added now to the discussion:

L514-18: 'However, investigations conducted in maize indicate that plant disease resistance is not a reliable predictor of the composition of the phyllosphere microbiota (*i.e.*, the microbial communities populating above-ground plant tissues), suggesting that the activity of host genetic determinants of the microbiota may be fine-tuned by organ-specific mechanisms.'

Reviewer #2

The authors investigated the microbiota of wild and domesticated barley, using metagenomics as an external phenotype. They identified a concise number of ASVs different between the parent lines. A segregating population between the two parents was generated, and their microbiome composition was correlated with genomic differences. One QTL region was identified that was robustly found at various taxonomic levels. To understand the function of the QTL, the authors generated sibling lines from a line heterogenous for this locus. The lines were similar in root morphology and exudation, but differed in microbiome composition and transcriptional profile. Again, only a concise number of genes was distinct between the lines as well as the sibling lines and the elite cultivar. Mapping of the differentially expressed genes pointed towards three genes found in the QTL region, one of them being a NLR gene likely involved in microbe perception and immune signaling. This candidate is close to a region important for yield. Selection for high yield in agriculture might thus have resulted in a distinct microbiota in barley cultivars.

The manuscript is well written, the figures well designed, and the conclusions well supported by data. The authors combine a number of recent technologies to identify host genes involved in shaping the microbiome. The approaches presented could be transferred to other plant species easily. The biological findings as well as the methods applied will be of interest to a broad readership.

Many thanks for the accurate summary and the positive evaluation! In the following section, we describe how we acted on the individual comments/suggestions.

Fig 1a: it seems that the microbiota are among others dominated by the large, gray ASV situated between elite and wild parents. what is that?

The ASV the reviewer is referring as a dominant one in Fig. 1a (ternary plot) was taxonomically classified as *Massilia* sp. Although this ASV is highly abundant in the rhizosphere, its position 'midway' in the baseline of the triangle indicates a comparable number of reads assigned to this ASV in both parental lines. Consistently, the differential enrichment analysis failed to identify this ASV among the differential significantly enriched features between the genotypes.

Line 108, 1b PCA: the segregant population seems to cluster around the elite parent, with quite some distance to the wild parent. Do you have an explanation for this?

This is an excellent observation! The most plausible explanation for this observation is that rhizosphere microbiota profiles mirror the host genetic composition. The segregating population was generated by 'crossing' the elite parent (*i.e.*, Barke) with the wild parent (*i.e.*, HID-144). The resulting F1 hybrid was further crossed with the elite parent (*i.e.*, a so-called backcross) prior giving rise to the segregating population via subsequent self-pollination (see PMID:25887319, Figure 1). Therefore, the population, for this reason designated BC₁S₃, contains ~50 % more elite genome than a population which would have been generated by a self-pollination of the initial F1 hybrid. This aspect is described in the result section of the submission:

L108-110: 'This distribution mirrored the increased proportion of elite genome expected in the original back-crossed BC₁S₃ population, with the majority of microbiota profiles of segregating individuals located spatially closer to the elite genotype (Fig. 1 b).'

supp table 4: please indicate number of roots analyzed.

We have indicated the number of root systems used per genotype (n=5) in Methods and Supplementary Table 4.

supp fig 4a: I am no expert at all on QTL analysis and thus my question likely has an obvious answer: isnt the heterozygous allele supposed to be a mix between elite and wild parents, and thus to be expected to localize at an intermediate level on the plots?

Not necessarily: in addition to additive effects (*i.e.*, the scenario depicted by the Reviewer) instances of dominance or even overdominance may occur, whereby the combined effect of two alternative alleles matches with one allele or exceeds their independent effects, respectively.

L236: please elaborate on the negative (soil) controls of this experiment. were metabolites not detected at all in pots without plants, or were levels significantly lower?

The metabolite concentrations in root exudates, unplanted controls and their differences has now been included in the revised version of Supplementary Database 2. In particular, we have added two spreadsheets to Supplementary Database 2 (Statistics polar compounds and Statistics non-polar compounds) with the statistical analyses of each of the metabolites in the unplanted controls and the root exudates of the different genotypes. We have eliminated three compounds (citric acid, L-Asparagine and L-Glutamine) that were found not significantly different from the unplanted controls compared to root exudates samples (Supplementary Figure 8). For clarity, we have also revised Supplementary Figure 7 and 8, rectifying the legend; and adding the 'unplanted control series' to the radar plots in Supplementary Figure 8. Details of the statistical analysis performed have now been included in L666-69.

L236: I assume in the GCMS experiment, some 100-1000 features were identified. were differences between the lines found when all features are taken into consideration?

The individual features identified can be found in Supplementary Database 2. The total number of compounds/features analysed was 55. We are aware that for plant extracts between 100 to 200 compounds can be identified (PMID:21148585), however the number of compounds we identified is comparable with the one of previous barley exudate characterisations (*e.g.*, PMID:29263712), possibly reflecting differences between the substrates (*i.e.*, root extract vs. root exudates).

In GC-MS electron ionisation is used, which generates multiple fragments of the parent ion. This means that for each compound we will generally get between 10-30 masses. If the reviewer considers features individual masses, then the total list of features will be much higher, however, we are only reporting individual compounds.

Finally, in data processing many automated data deconvoluting and processing software tools (e.g., XCMS) often generate outputs in the hundreds or thousands of features. In this report, we have done peak picking and identification manually which in turn will likely lead to less features being captured (particularly in low abundance peaks), however, this approach ensures that false positive components (or artefacts from processing) are minimised.

L455: typo: hypothesize

Corrected

Reviewer #3

Plant and its surrounding microorganisms form complex networks which is important for plant adaptation to the environments. It is well known that different plants have distinguish microbiota and different crop production systems also have fundamental impact on the microbiota in the rhizosphere. It is much unknown how plant genes shape the microbiota, which is critical for genetic improvement of crops for environmental adaptation. This paper first used metagenomics information as an external quantitative phenotype to map the genetic loci to determine microbiota profiles and then identify the quantitative loci and candidate genes. It would be ideal if the authors can confirm some of the genes' functions. However, this is not essential for the paper to be accepted for publication. The experimental designs and data analysis are robust and the results are well supported by the experimental data. I have only a few minor suggestions for the authors.

Many thanks for the accurate summary and the positive evaluation! We concur with the reviewer on the importance of the validation/functional characterisation of individual candidate genes. This effort will be indeed centre stage in future lines of investigations of the lab. Likewise, as

this effort will require significant time and resources, we share the reviewer's opinion that said characterisation appears beyond the scope of the presented investigation.

1. The result for QTL mapping is too simple. The authors should provide more information for Fig 2. The current presentation did not reflect the complexity of plant and microbiota interaction.

We are aware that Figure 2 represents a reductionistic approach as, for example, may not recapitulate additional features of the microbiota (e.g., microbe-microbe interactions). However, it is a “classical” graphic representation used frequently in QTL analysis to depict genetic associations. Furthermore, investigations on differentially enriched members of the microbiota represent a common way to characterise the so-called host genotype effect. We therefore reasoned this type of graphic representation would benefit both geneticists and microbial ecologists. We have now rectified the title of the caption to better reflect this approach (*i.e.*, now reads ‘...individual bacterial members of the rhizosphere microbiota’ instead of ‘...rhizosphere bacterial microbiota’ as a whole) and revised the sentence introducing Figure 2 as follows:

L113-17 ‘To gain novel insights into the host genetic control of the rhizosphere microbiota, we developed a reductionistic approach whereby we used taxa that were differentially recruited between the parental lines and their abundances in the segregating population as “quantitative phenotypes” to search for significant associations with genetic markers located throughout the barley genome.’

The authors over emphasized NLR gene and biotic stress tolerance in the discussion and other parts of the paper. I suggest that the authors consider a balance presentation of the plant genes for both biotic and abiotic stress tolerances and adaptation. The abiotic stress and plant interaction may play more important role to shape the microbiota in rhizosphere. There are some reports that xyloglucan endotransglucosylase/hydrolase enzyme is involved in abiotic stress responses, e.g. drought tolerance. Thus, the discussion part need expand the discussion on this gene.

Many thanks for this suggestion. The emphasis on the *NLR* gene is dictated also by the fact that this candidate maps in an area of structural variation of the barley genome, a distinctive feature of the locus. Having said that, we have acted on this comment by further elaborating on

references illustrating the xyloglucan endotransglucosylase/hydrolase enzyme function during abiotic stresses and its potential role in shaping the rhizosphere microbiota.

L492-98: 'An additional contribution of *XTH* genes to host-microbiota interactions may be represented by an increased adaptation to soil chemical and physical conditions. For instance, *XTH* genes have previously been implicated in abiotic stress tolerance, including drought, salt stress and cold acclimation. As wild barley accessions have evolved under marginal soil conditions, these may have imposed a selective pressure on the genetic diversity of *XTH* genes, which, in turn, may have led to a differential microbial recruitment.'

The pan genome analysis is good to identify some structural variation in the QTL region, but without the genome sequence of the wild barley donor, there is limitation for the analysis, e.g. missing some key genes in the QTL region based on the other genomes sequences.

This is a valid point. We are aware that our colleagues at the Martin Luther University Halle-Wittenberg (Germany) are undertaking the effort of generating reference-quality genomes for some of the wild parental lines of the HEB-25 population (e.g., PMID: 25887319) used in this study (<https://gepris.dfg.de/gepris/projekt/433162815?language=en>), although those are not yet accessible for experimentation. While the lack of a wild genome may be perceived as a limitation of the study, it is important to note that the approach we adopted, and the candidate genes identified, will retain their scientific significance regardless of the genetic diversity harboured by wild accessions at the locus. To acknowledge this scenario, we have introduced an additional sentence in the concluding section of the manuscript which reads:

L531-34: 'Our approach can be readily used to identify other or additional candidate genes from reference-quality genomes, including wild ancestors, as they become available for experimentation, in barley and other species'.

There are some minor spelling or grammar errors, e.g. allele vs alleles

We rectified typos throughout the text.

#END

Reviewers' Comments:

Reviewer #1:

Remarks to the Author:

The authors have done an exceptional job addressing my comments on the original manuscript. I have no further suggestions. Congratulations on an excellent paper

Reviewer #2:

Remarks to the Author:

Dear authors,

thank you for addressing the issues raised in the first review process. I am happy with how my points were integrated into the new version of the manuscript, and I find the manuscript much improved now.

Reviewer #3:

Remarks to the Author:

I have reviewed the previous version of the paper. My comments and suggestions are now well addressed by the authors.

Point-to-point response letter for the manuscript 'Identifying Plant Genes Shaping Microbiota Composition In The Barley Rhizosphere'

As no formal request emerged from the last iteration of the review, we would like to use this letter to thank once again the three Reviewers for their time and the insightful and constructive comments on our submission.

#END